# Measurement Report: Investigation of pH- and particle size-dependent chemical and optical properties of water-soluble organic carbon: implications for its sources and aging processes

Yuanyuan Qin[1], Juanjuan Qin[1,2], Xiaobo Wang[1], Kang Xiao[1], Ting Qi[1,6], Yuwei Gao[1], Xueming Zhou[5], Shaoxuan Shi[1], Jingnan Li[1], Jingsi Gao[3], Ziyin Zhang[4], Jihua Tan[1], Yang Zhang[1], and Rongzhi Chen[1]

[1] College of Resources and Environment, University of Chinese Academy of Sciences, Beijing, 100049, China

[2] Guangzhou Institute of Geochemistry, Chinese Academy of Sciences, Guangzhou, 510640, China

[3] School of Materials & Environmental Engineering, Shenzhen Polytechnic, Shenzhen, 518000, China

[4] Institute of Urban Meteorology, China Meteorological Administration, Beijing, 100089, China

[5] Faculty of Earth Resources, China University of Geosciences, Wuhan, 430074, China

[6] School of Chemical Sciences, University of Chinese Academy of Sciences, Beijing, 100049, China

*Correspondence*: Jihua Tan (tanjh@ucas.ac.cn) and Yang Zhang (zhangyang@ucas.ac.cn)

**Abstract.** Knowledge of the chemical structures and optical properties of water-soluble organic carbon (WSOC) is critical considering its involvement in many key aerosol-associated chemical reactions and its potential impacts on climate radiative forcing. This study investigates the coupled effects of pH and particle size on the chemical structures (functional groups) and optical properties (UV/fluorescence properties) of WSOC and to further explore the aging and source of WSOC constituents. The results showed that the specific UV absorbance at a wavelength of 254 nm ($SUVA_{254}$) and mass absorption efficiency at a wavelength of 365 nm ($MAE_{365}$) were higher in smaller than larger particles, revealing the relatively higher aromaticity/molecular weight and more freshness of WSOC in smaller particles. A decrease in aromaticity/molecular weight of WSOC in larger particles was caused by the degradation reaction that occurred during the aging process. The carboxylic groups tend to be enriched in larger particles, whereas the contribution of phenolic groups was generally higher in smaller particles. The changes in the fluorescence peak position suggested that hydroxyl groups play a leading role in pH-responsive fluorescence in summer, while carboxylic and nitro groups play a dominant role in winter. Overall, the chromophores in smaller particles showed a more pronounced pH dependence, which might be related to the higher content of aromatic species in WSOC in these particle size ranges. Specifically, the climate impact of WSOC would be enhanced with increasing pH. The

pH- and particle size-dependent chemical and optical properties of WSOC provide insights into the structure, aging, and source of WSOC, which will ultimately improve the accuracy of assessing the climate effects of WSOC.

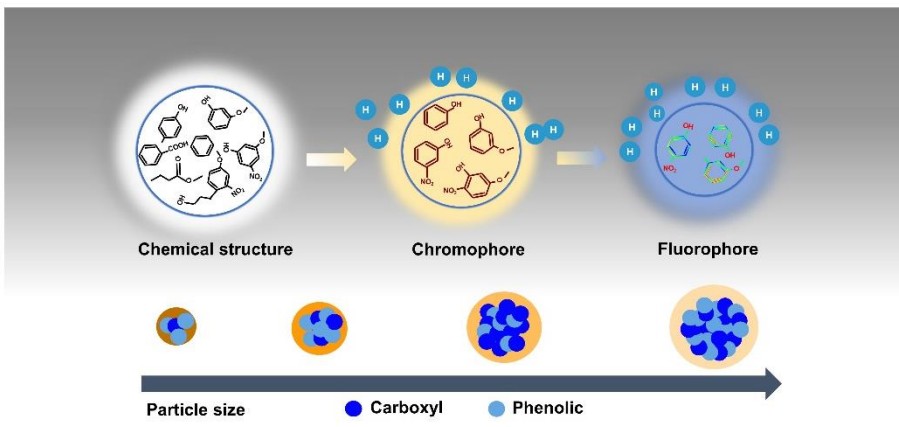

30

**Graphical abstract**

35

# 1 Introduction

Water-soluble organic carbon (WSOC, see Appendix A for a list of abbreviations) comprises a considerable fraction of organic aerosol mass (10%–80%) (Horník et al., 2021), and plays important roles on climate change (Chen et al., 2020; Sun et al., 2011) and air quality (Snyder et al., 2009). WSOC is released from anthropogenic (e.g., biomass burning and coal combustion) and natural sources, and can also be formed through complex secondary reactions (Yu et al., 2017; Wu et al., 2018). The sources of WSOC vary significantly with location and season, e.g., WSOC in $PM_{2.5}$ was primarily derived from secondary aerosol formation and biomass burning in Korea (Park et al., 2015), and from coal combustion, biogenic emission, and secondary aerosol formation in winter and biogenic emission and secondary oxygenation of vehicle exhaust in summer in a northwest city of China (Qin et al., 2018).

The formation and transformation of WSOC in atmospheric particulate matter is highly complex. Existing studies mostly focused on investigating the concentration levels and optical and chemical characteristics of bulk WSOC in $PM_{2.5}$ (Xiang et al., 2017; Ma et al., 2022). Studies focusing on size-resolved WSOC are still limited, e.g., earlier studies focused on exploring the size distribution of WSOC (Timonen et al., 2008; Deshmukh et al., 2016), and a few recent studies investigated the optical properties of size-resolved WSOC (Chen et al., 2019; Qin et al., 2021a). The sources, formation mechanisms, and transformation processes of WSOC are strongly related to particle size distribution (Chen et al., 2019). Therefore, the size-distribution of WSOC can serve as a good indicator of its sources, fate, and aging processes (Boreddy et al., 2021; Jang et al., 2019; Frka et al., 2018). For example, Frka et al. (2018) found that wood burning was the most important source of humic-like substances (HULIS) in the aerosol accumulation mode (from ~0.1 to ~2 μm) during the autumn and winter; Jang et al. (2019) reported that HULIS in smaller particles was likely derived from local sources, while in larger particles from secondary organic aerosols (SOA) in the atmosphere, and Qin et al. (2021a) found that the fluorescence properties of WSOC varied with the particle size, and the fluorescence characteristics of different particle sizes could be used to reveal the aging of WSOC. To date, a knowledge gap still remains regarding the aging, fate, and important atmospheric processes modulating the chemical and physical properties of WSOC due to the dearth of WSOC-focused research in different particle sizes and the limitations of analytical methods.

However, only investigating the chemical and optical properties of WSOC cannot fully understand its formation and

transformation processes because environmental conditions, such as relative humidity (RH), solar radiation intensity, and temperature, can affect the physical and chemical properties of aerosol particles (Bousiotis et al., 2021; Ge et al., 2021). Besides, aerosol acidity (pH) affects the formation of secondary organic aerosols (SOA) via altering chemical reaction pathways (Ault, 2020) and ultimately affect the global climate. It should be noted that aerosol pH increased with increasing particle size (Craig et al., 2018). Due to the complexity of the physicochemical properties of size-resolved aerosols, our understanding of their pH and pH-dependent behavior is still lacking. Examining pH-dependent chemical structures and optical properties of size-resolved WSOC would be a worthy attempt to help reveal their formation mechanisms.

The commonly used analytical methods to characterize the optical properties of WSOC are three-dimensional excitation-emission matrix (EEM) spectroscopy and ultraviolet–visible (UV–Vis) absorption spectroscopy (Zhang et al., 2021; Yang et al., 2020). The EEM spectroscopy is a rapid as well as informative method to identify chromophores that may not be distinguished by UV–Vis absorption spectroscopy (Chen et al., 2019; Xiao et al., 2020). Therefore, EEM spectroscopy has been widely applied in atmospheric WSOC characterization (Fu et al., 2015; Qin et al., 2018). However, such a technique has not been widely applied to investigate the fluorescence properties of WSOC in different particle sizes. Fourier transform infrared (FTIR) spectroscopy has been frequently used for the identification of WSOC functional groups (Chen et al., 2016a), although this analysis is difficult to perform quantitatively. pH titration enables qualitative and quantitative analyses of functional groups on the surface of substances (Zhang et al., 2011; Xiao et al., 2014), and this approach has recently been successfully applied to the characterization of WSOC in ambient $PM_{2.5}$ (Qin et al., 2021b), but not yet on size-resolved WSOCs.

To better understand the structure, source, and aging of WSOC, this study explored the optical properties and chemical structure from the perspective of pH and particle size response. Firstly, the optical properties of four WSOC with representative particle sizes of < 0.26, 0.44–0.77, 1.40–2.50, and 2.50–10.0 μm were determined with a UV–Vis absorption spectroscopy and EEM fluorescence spectroscopy combined with parallel factor analysis (PARAFAC). Then, the functional groups of WSOC were analyzed using a FTIR spectroscopy and pH titration. Subsequently, the influence of pH on the optical properties of WSOC in different particle sizes was examined. Finally, the environmental implications of pH-dependent and particle size-dependent behaviors of WSOC were discussed.

## 2 Experimental methods

### 2.1 Sample collection and preparation

#### 2.1.1 Sample collection

Particulate samples were collected on the roof of a building (~20 m above the ground) inside the campus of the University of Chinese Academy of Sciences (40°24′N, 116°40′E) in Huairou District of Beijing, China. The sampling site is in a typical urban environment surrounded by schools, research institutes, and hospitals. There were no obvious industrial sources nearby, and pollutants were mainly derived from regional transport. The samples were collected on prebaked (under 550℃ for 5.5 h) quartz fiber filters (Φ 90 mm, Whatman) using a 6-stage micro-orifice uniform deposit impactor (MOUDI), with aerodynamic

cut-point diameters of 0.26, 0.44, 0.77, 1.40, 2.50, and 10.0 μm. A total of 46 sets of six-stage size-segregated aerosol samples were collected during two summer and one winter months from June 2019 to August 2020, with 35 sets of six-stage size-segregated aerosol samples in summer and 11 sets of six-stage size-segregated aerosol samples in winter. To collect sufficient mass of particulate matter in each sample, the sampling duration for each individual sample was selected based on the degree of air pollution with a minimum of 1 day on polluted days to a maximum of 7 days on clean days, but all the sampling started

at from 8:00 a.m. and ended at 7:00 a.m in 1-7 days. The collected samples were then stored at −20℃ until further analysis.

#### 2.1.2 WSOC extraction

A quarter of all filters of each size (summer/winter) were mixed together in a bottle, extracted twice via ultrasonication in Milli-Q water for 15 min to achieve the extensive release of solubilized WSOC, and then the extracted liquid was filtered through a 0.22 μm membrane filter to remove insoluble suspensions. Blank filters, used for background checks, were also

extracted under the same conditions. The average WSOC concentration of the blank filter was 0.39 mg L$^{-1}$.

#### 2.1.3 pH titration

HCl and NaOH were used to adjust the pH of the WSOC solutions. The pH of the WSOC for UV–Vis absorption and EEM fluorescence spectrometry was controlled in the range of 2–10 at 1 unit interval and recorded using a pH meter (Mettler Toledo, Swiss). The pH meter was calibrated before running any titration. The raw pH of WSOC with particle sizes of < 0.26 μm,

0.44–0.77 μm, 1.40–2.50 μm, and 2.50–10.0 μm was 5.61, 5.75, 5.89, and 6.19, respectively, in summer, and 6.19, 6.49, 6.64, and 6.83, respectively, in winter.

The distribution of acidic groups was also obtained by pH titration as described in detail elsewhere (Wang and Waite, 2009). Briefly, prior to titration, a 50 mL WSOC solution was first acidified to a pH below 3 by addition of HCl solution, and then the titration was performed until the pH was higher than 10 by stepwise addition of 0.1 M NaOH. Throughout the titration, the

115 WSOC solution was bubbled with pure $N_2$ to remove $CO_2$ from the air.

## 2.2 Instrumental analysis

### 2.2.1 Total organic carbon (TOC) and FTIR analysis

The WSOC concentration was quantified by a TOC analyzer (Analytic Jena AG multi N/C3100, Germany). Prior to measurement, a drop of 2 mol $L^{-1}$ HCl was added to the WSOC solution to remove the interference of inorganic carbon.

A Perkin Elmer FTIR (Frontier) spectrometer was employed to investigate the functional groups of WSOC. A fully dried mixture of the lyophilized WSOC and KBr was ground in an agate mortar and pressed into discs for FTIR analysis. FTIR spectra were recorded in the range of 4000–400 $cm^{-1}$ with a 1 $cm^{-1}$ interval at a spectral resolution of 4 $cm^{-1}$. Pure KBr was measured under the same conditions and its spectra was subtracted from the sample spectra for background correction.

### 2.2.2 UV–Vis absorption and EEM fluorescence spectra

UV–Vis absorption and EEM fluorescence spectra of WSOC were recorded in a 1 cm path-length quartz cell using a UV–visible spectrophotometer (UV–2401PC, Shimadzu, Japan) and a fluorescence spectrophotometer (Agilent Cary Eclipse, America), respectively. The UV–Vis absorption spectra for all samples were measured over the wavelength range from 200 to 500 nm with an interval of 1 nm. The EEM fluorescence spectra were recorded in the wavelength range of 200 to 400 nm for excitation and 250 to 500 nm for emission with an interval of 5 nm.

## 2.3 Data analysis

### 2.3.1 Acidic group distributions

The distribution of acidic groups ($pK_a$ and group density) was calculated from the pH titration data using linear programming

optimization as follows (Wang and Waite, 2009):

$$\sum C_j\alpha_{ij} - C_{ANC} = [base] - [acid] - [carb] + [H^+] - [OH^-] \,, \tag{1}$$

$$\alpha_{ij} = \frac{K_{aj}}{K_{aj}+[H^+]} \,, \tag{2}$$

where the term $\sum C_j\alpha_{ij}$ is the sum of unreacted functional groups at each titration step ($i = 1$, m), $C_j$ denotes the concentration

of the $j^{th}$ functional group, $K_{aj}$ ($j=1$, n) represents the conditional dissociation constants ($K_{aj}=10^{-pKaj}$), and $C_{ANC}$ denotes the

acid-neutralizing capacity of the system (defined as the sum of all non-reacting cations minus non-reacting anions). The

concentrations of acid ([acid]) and carbonate ([carb]=[$HCO_3^-$]+2[$CO_3^{2-}$]) were zero in this study since the titration starting pH

was below 3.

### 2.3.2 UV–Vis absorption spectra

Based on the UV–Vis absorption spectra, the specific UV absorbance at a wavelength of 254 nm (SUVA$_{254}$, m$^2$ g$^{-1}$), mass

absorption efficiency (MAE$_\lambda$, m$^2$ g$^{-1}$), Absorption Ångström Exponent (AAE), and the difference absorbance spectra

($\Delta$absorbance ($\lambda$), m$^2$ g$^{-1}$) were calculated according to the following Eqs. (Jane et al., 2017):

$$\text{SUVA}_{254} = \frac{A_{254}}{C \times L} \,, \tag{3}$$

$$\text{MAE}_\lambda = \frac{A_\lambda}{C \times L} \times \ln(10) \,, \tag{4}$$

$$\text{MAE}_\lambda = K \times \lambda^{-AAE} \; (330 \text{ nm} \leq \lambda \leq 400 \text{ nm}) \,, \tag{5}$$

$$\Delta\text{absorbance }(\lambda) = \frac{A(\lambda)_{pH} - A(\lambda)_{ref}}{C \times L} \,, \tag{6}$$

where $A_\lambda$ denotes the absorbance at wavelength $\lambda$, $C$ is the mass concentration of WSOC, $L$ is the cell path length (1 cm), and

$K$ is a constant related to light absorption. Additionally, the $A(\lambda)_{pH}$ and $A(\lambda)_{ref}$ are the absorption spectra at a different pH and

at a reference pH (2 and 7), respectively.

### 2.3.3 EEM and PARAFAC analysis

The raw EEM spectra were processed using the following procedure described in detail elsewhere (Xiao et al., 2018b). The

pure water was subtracted from the EEM spectra of WSOC as background correction. The interfering fluorescence signals of

the Rayleigh and Raman scattering were then eliminated by an interpolation technique. The UV–Vis absorbance in the wavelength range of 200–500 nm was used to correct the inner-filter effect of fluorescence intensity. Subsequently, the fluorescence intensity was normalized to Raman units (RU) using the Raman peak area of pure water, and was further divided by the TOC concentration to obtain the specific fluorescence intensity per unit TOC (SFI). Detailed information on EEM fluorescence properties, such as average fluorescence intensity per unit TOC at each emission wavelength ($FI_m$/TOC), apparent quantum yield (AQY), and Stokes shift, were further extracted. AQY is defined as the ratio of the number of emitted photons to the number of absorbed photons after the fluorophores absorbing light (Xiao et al., 2018b). The Stokes shift, is defined as the difference between the excitation and emission wavenumbers, and can be calculated as follows:

$$SFI = \frac{FI}{TOC} = \frac{1}{TOC}\left(\frac{1}{N}\sum_{Ex}\sum_{Em} l\right), \tag{7}$$

$$\frac{FI_m}{TOC} = \frac{1}{TOC}\left(\frac{1}{N_{Ex}}\sum_{Ex} l\right), \tag{8}$$

$$AQY = \frac{\int_{Em} FI(\lambda_{Ex}, \lambda_{Em})\mathrm{d}\lambda_{Em}}{UVA(\lambda_{Ex})\int_{Em}\mathrm{d}\lambda_{Em}}\bigg|_{Ex}, \tag{9}$$

$$Stokes\ shift = \frac{1}{\lambda_{Ex}} - \frac{1}{\lambda_{Em}}, \tag{10}$$

where FI is the fluorescence intensity, $I$ is the fluorescence intensity at each Ex/Em wavelength position, $N$ is the total number of EEM data, $N_{Ex}$ denotes the total number of data under each Ex wavelength position, UVA is the average absorbance, and $\lambda_{Ex}$ and $\lambda_{Em}$ are the excitation and emission wavelengths (nm), respectively.

The different independent fluorescent components were identified by PARAFAC analysis using the DOM-Fluor toolbox (Wang et al., 2022). Three independent components were acquired based on split half analysis, residual analysis, and visual inspection (Wu et al., 2011). Additionally, the independent-samples $t$-test was used to evaluate the seasonal differences in light absorption and fluorescence properties of WSOC.

## 2.4. Quality assurance and quality control

Quality assurance and quality control (QA/QC) procedures were applied through the laboratory and instrumental analysis processes. Before sampling, all quartz fiber filters were baked at 550°C for 5.5 h. Prior to analysis, all vials were washed with Milli-Q water, dried, and preheated at 550°C for 5.5 h in a muffle furnace, and then wrapped in aluminum foil before being

used. WSOC extraction and measurement were conducted in a designated laboratory. All WSOC concentrations, UV–Vis absorption spectra, EEM fluorescence spectra, and functional groups data were blank-corrected. All instruments operated in this study required regular calibration. Sterile gloves and robes were worn throughout the experimental procedure to avoid contamination and to ensure the accuracy of the test results.

## 3 Results and discussion

### 3.1 Optical properties and chemical structures of WSOC

### 3.1.1 Optical properties at raw pH

The light absorption and fluorescence properties of WSOC varied with particle size (Table 1). Higher $SUVA_{254}$ for particles of < 0.26 μm than other particle sizes highlighted the relatively higher aromaticity/molecular weight of WSOC in smaller particles (Cawley et al., 2013). Baduel et al. (2011) found that HULIS undergoes degradation under UV and ozone conditions, which in turn reduces its aromaticity and molecular weight. Thus, the higher $SUVA_{254}$ values in smaller particles in this study may be due to fresh WSOC in these particle size ranges. In contrast, WSOC in larger particles undergoes a series of degradation reactions during the aging process, resulting in a decrease in aromaticity/molecular weight. The $MAE_{365}$ values of particles in the size range of <0.26 μm, 0.44–0.77 μm, 1.40–2.50 μm, and 2.50–10.0 μm were 0.1258, 0.1321, 0.1014, and 0.1145 $m^2$ $g^{-1}$ in summer, respectively, and 1.2615, 0.7991, 0.8206, and 0.3707 $m^2$ $g^{-1}$ in winter, respectively, indicating that WSOC in smaller particles had stronger light absorption capabilities (Huang et al., 2022). This is because WSOC in smaller particles contained more chromophores, such as nitrogen chromophores (see Section 3.1.2), resulting in stronger light absorption capabilities (Wu et al., 2018). The average AAE values were the highest in particle sizes of <0.26 μm, suggesting the existence of more wavelength dependence light absorption properties in smaller particles (Wu et al., 2018).

**Table 1.** The optical properties of WSOC in particles of different size ranges (μm) at raw pH.

| | < 0.26 μm | | 0.44–0.77 μm | | 1.40–2.50 μm | | 2.50–10.0 μm | |
| --- | --- | --- | --- | --- | --- | --- | --- | --- |
| | Summer | Winter | Summer | Winter | Summer | Winter | Summer | Winter |
| $SUVA_{254}$ ($m^2\,g^{-1}$) | 0.0064 | 0.0317 | 0.0054 | 0.0237 | 0.0062 | 0.0181 | 0.0063 | 0.0093 |
| $MAE_{365}$ ($m^2\,g^{-1}$) | 0.1258 | 1.2615 | 0.1321 | 0.7991 | 0.1014 | 0.8206 | 0.1145 | 0.3707 |
| AAE | 9.1573 | 5.4345 | 6.7218 | 6.3975 | 10.047 | 4.3854 | 5.2922 | 4.2987 |
| FI/TOC (R.U. $mg^{-1}$ L) | 0.0644 | 0.6973 | 0.0432 | 0.4440 | 0.0487 | 0.2408 | 0.0133 | 0.0990 |
| AQY | 0.2967 | 0.6261 | 0.2555 | 0.5993 | 0.2644 | 0.4242 | 0.0506 | 0.4826 |
| Stokes shift ($μm^{-1}$) | 0.0105 | 0.0173 | 0.0119 | 0.0190 | 0.0212 | 0.0154 | 0.0121 | 0.0128 |

The overall fluorescence intensity per TOC (FI/TOC) decreased steadily with increasing particle size (Table 1). The fluorescence intensity was higher for fresh than aged brown carbon (BrC), as previously observed by Fan et al. (2020). Therefore, the FI/TOC further highlights that WSOC may have undergone a cascade of aging processes (e.g., photochemical aging and oxidative aging) with increasing particle size, and thus has weaken its fluorescence intensity (Kuang et al., 2021; Wu et al., 2021). AQY exhibited a similar trend to FI/TOC, indicating that a large scale of the π-conjugated system and less electron-withdrawing groups (e.g. $-NH_3^+$ and $-COOH$) seemed to be present in smaller particles (Xiao et al., 2020; Xiao et al., 2018b). For summer WSOC, the Stokes shift of the particle size of 1.40–2.50 μm was higher than that of the other particle sizes, indicating greater energy loss due to relaxation in the excited states of the fluorophores in larger particles. For winter WSOC, however, the particle sizes of 0.44–0.77 μm showed an even higher Stokes shift.

It is noteworthy that the light absorption and fluorescence properties of WSOC exhibited strong seasonal variations, with higher $SUVA_{254}$ and $MAE_{365}$ values in winter than in summer. This suggests higher aromaticity/molecular weight and more chromophores of WSOC in winter. The different sources and/or formation processes associated with different weather conditions (e.g., temperature and relative humidity) between the two seasons likely caused the seasonal variations in WSOC properties mentioned above. Literature reported $MAE_{365}$ from different sources showed values of 0.76–2.47 $m^2\,g^{-1}$ for biomass burning smoke, 0.20–1.33 $m^2\,g^{-1}$ for fossil fuel combustion, and -0.11±0.20 $m^2\,g^{-1}$ for biogenic SOA (Geng et al., 2020). Our previous study also found higher $MAE_{365}$ from combustion sources than ambient samples (Qin et al., 2022). The $MAE_{365}$ values observed in the present study suggest that WSOC may be mainly derived from secondary formation in summer and

from mixed sources of primary emissions (e.g., biomass burning and fossil fuel combustion) in winter. The FI/TOC values also exhibited strong seasonal variations with 4−10 times higher values in winter than in summer, indicating enhanced photooxidation or photobleaching reactions of WSOC in summer due to stronger UV light. Furthermore, there are significant seasonal variations in the optical properties (e.g., $SUVA_{254}$, $MAE_{365}$, and FI/TOC) of WSOC in smaller particles ($t$-test, $p$=0.021), compared to the relatively flat seasonal patterns in larger particles, further suggesting significant seasonal differences in the sources and chemical structures of WSOC in smaller particles.

### 3.1.2 Functional groups analysis at raw pH

The FTIR spectra of WSOC in particles of different sizes are presented in Fig. 1. The FTIR spectra predominantly exhibited the presence of oxygen containing functional groups and aliphatic C–H groups for all samples (Duarte et al., 2005). Strong absorptions of C−OH (3429 cm$^{-1}$) were observed for all samples, indicating that WSOC contained abundant phenol, hydroxyl, and carboxyl groups (Duarte et al., 2005). A weak absorption at 3175 cm$^{-1}$ was observed in both < 0.26 μm and 0.44–0.77 μm samples, typically corresponding to the N–H stretching vibration of amines and amides (Huo et al., 2008), but this feature was not shown in the spectra of the large particles, indicating a higher amide content in smaller particles. In addition, an absorption peak at 1720 cm$^{-1}$ was only present in the smallest particles with sizes of < 0.26 μm in summer, which was attributed to the unconjugated C=O stretching mainly of carbonyl carbon (Hu et al., 2019). A peak at 1641 cm$^{-1}$ was also previously reported, which was attributed to conjugated carbonyl (C=O) groups and aromatic rings (C=C) (Zhang et al., 2022). A strong and sharp absorption at 1389 cm$^{-1}$ is usually attributed to the C–H asymmetric bending vibrations of methyl groups in aliphatic chains (Duarte et al., 2007; Colthup, 2012). These results suggest that WSOC of different sizes contain abundant branched structures. The presence of a peak at 1114 cm$^{-1}$ typically corresponded to the stretching vibration of C–OH, mainly alcohol (Chen et al., 2017). Strong absorptions of the out-of-plane vibrations of C–H (625 cm$^{-1}$) and C=C (827 cm$^{-1}$) groups were observed in the spectra, indicating that WSOC also contained abundant alkenes (Duarte et al., 2007). It is obvious that smaller particles contained more oxygenated species and nitrogenous organic compounds than larger particles did. Nitrogenous organic compounds with different aromatic structures are considered to be important light absorbers (Wu et al., 2022), resulting in stronger light absorption of WSOC in smaller particles. This is consistent with the results from the optical absorption properties

analysis discussed in Section 3.1.1.

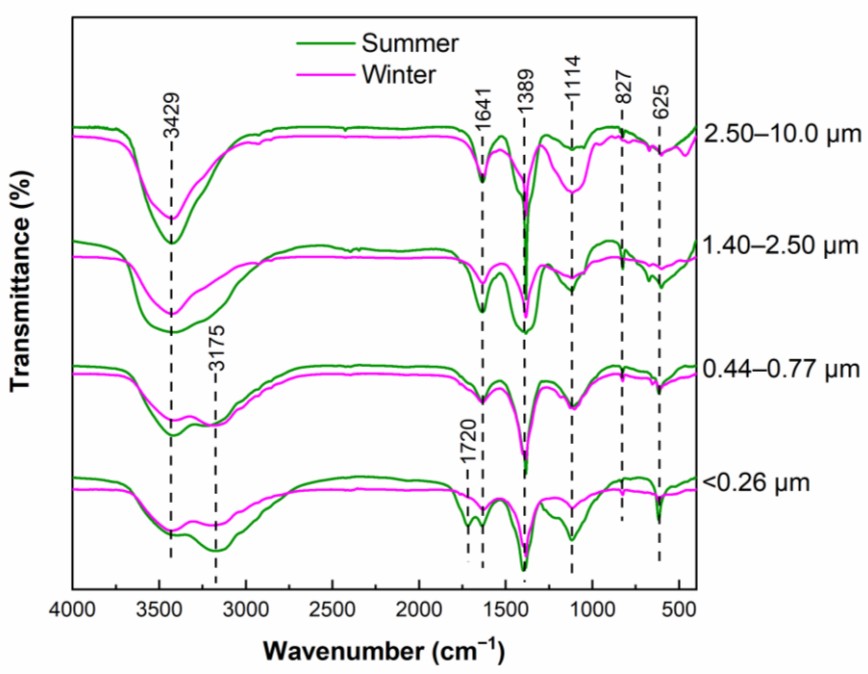

**Figure 1.** FTIR spectra of WSOC in particles of different sizes in summer and winter at raw pH.

### 3.1.3 Acidic group distributions

Figure 2 shows the distributions of the acidic groups of WSOC with a $pK_a$ in the range of 3.0–9.0. The $pK_a$ values of carboxylic and phenolic groups are in the range of 3.5–5.6 and 8.1–9.0, respectively, and the $pK_a$ range of 6.3–7.9 may be an overlapping region among weak carboxylic groups, phosphoric acids or phenols (Mu et al., 2019). The contribution of (strong) carboxylic groups showed a clear characteristic of particle size distribution, with the highest percentage in sizes of 1.40–2.50 μm and the lowest one in sizes of < 0.26 μm, reflecting that (strong) carboxylic groups tend to exist in larger particles, a phenomenon that is consistent with the findings in AQY mentioned above. In contrast, the contribution of (strong) phenolic groups was the highest in smaller particles (< 0.77 μm) and the lowest in larger particles (1.40–2.50 μm). This pattern indicates that the phenolic groups were abundant in WSOC of smaller particles. It has been demonstrated that aromatic compounds (e.g., phenol) are abundant in biomass burning particles (Laskin et al., 2015; Lin et al., 2016; Sannigrahi et al., 2006). Additionally, nitrophenols and their derivatives have been found to be possibly associated with the gas-phase oxidation of anthropogenic

VOCs and aqueous-phase oxidation processes in polluted high-NO$_x$ environments (Frka et al., 2022; Wang et al., 2019). Therefore, the size-distribution of the acidic group clearly indicates that WSOC with sizes of < 0.77 µm mainly originated from biomass burning, although the contribution of secondary formation should not be completely neglected. Overall, the contribution of strong phenolic groups (>54%) was higher than that of carboxyl groups in all samples except particle sizes of 1.40–2.50 µm. This result indicates that phenolic groups were more abundant than carboxylic groups in WSOC. Meanwhile, except for particle sizes of 1.40–2.50 µm, the contribution of carboxyl was higher and the phenolic group was lower in summer than in winter.

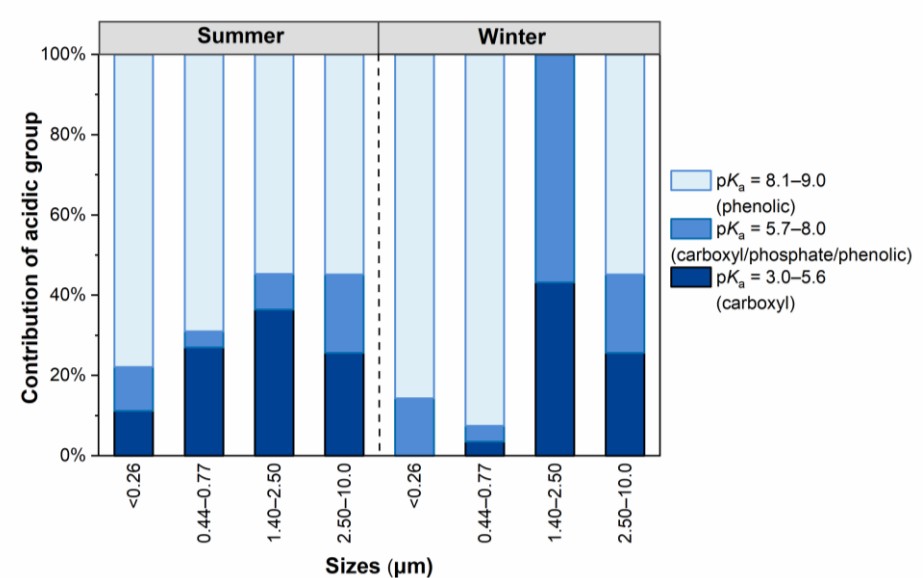

**Figure 2.** The distribution of the acidic group of WSOC in particles of different sizes in summer and winter.

### 3.2 Role of pH on UV–Vis absorption spectra

### 3.2.1 Absorption spectra

In addition to the particle size, the aerosol pH is another factor affecting the structure of WSOC chromophores. As shown in Fig. 3, the absorption of WSOC gradually increased with increasing pH for almost all samples, which may be related to the deprotonation of aromatic WSOC chromophores (Korshin et al., 1997; Young et al., 2018). On average, the absorbance for particle sizes of < 0.26 µm, 0.44–0.77 µm, 1.40–2.50 µm, and 2.50–10.0 µm increased by 4.6 %, 1.3 %, 0.6 %, and 0.9 %,

respectively, per unit pH increase in summer, and by 1.3 %, 0.5 %, 0.5 %, and 2.9 %, respectively, in winter. The results suggest that the absorption in particle sizes of < 0.26 μm and 0.44–0.77 μm in summer and < 0.26 μm and 2.50–10.0 μm in winter showed a more pronounced pH dependence, as further illustrated below.

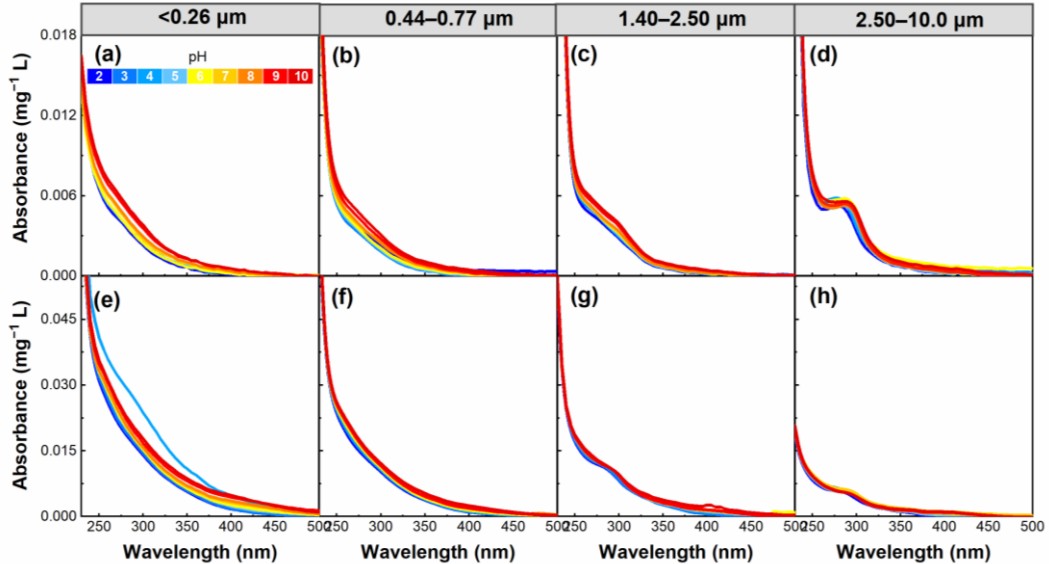

**Figure 3.** pH dependence of absorption per unit mass of WSOC in particles of different sizes in summer (upper row) and winter (lower row) in the pH range of 2–10.

### 3.2.2 Mass absorption efficiency at 365 nm (MAE$_{365}$)

The MAE$_{365}$ values showed a clear trend, that is, an overall increase in MAE$_{365}$ was observed with increasing pH (Fig. 4), consistent with the results of a previous study (Mo et al., 2017). This finding indicates that WSOC had a stronger light absorption capability at high pH. However, the MAE$_{365}$ for particles < 0.26 μm in winter, exhibited the highest value at pH 4, most likely because WSOC of particles in this size range were significantly influenced by the group of p$K_a$ values near this pH value. The pH-dependent MAE$_{365}$ suggests that under different pH conditions WSOC may have a different impact on climate

(i.e., climate impact would be enhanced as pH increases) (Aiona et al., 2018). Overall, the average MAE$_{365}$ for particles of < 0.26 μm changed more sharply with increasing pH (with a 12.7% increase per pH unit) due to the stronger pH-dependence of the absorption spectra of smaller particles compared to the cases of larger particles, as described above. Phillips et al. (2017)

found that the variations of the light absorption properties of BrC with pH were the result of structural changes in the nitro-aromatics and phenols. More aromatic species were found in the smaller than larger particles in this study (Table 1). Therefore, the strong dependence of light absorption properties on pH in smaller particles might be related to the higher content of aromatic species (e.g., nitro-aromatic species) in WSOC.

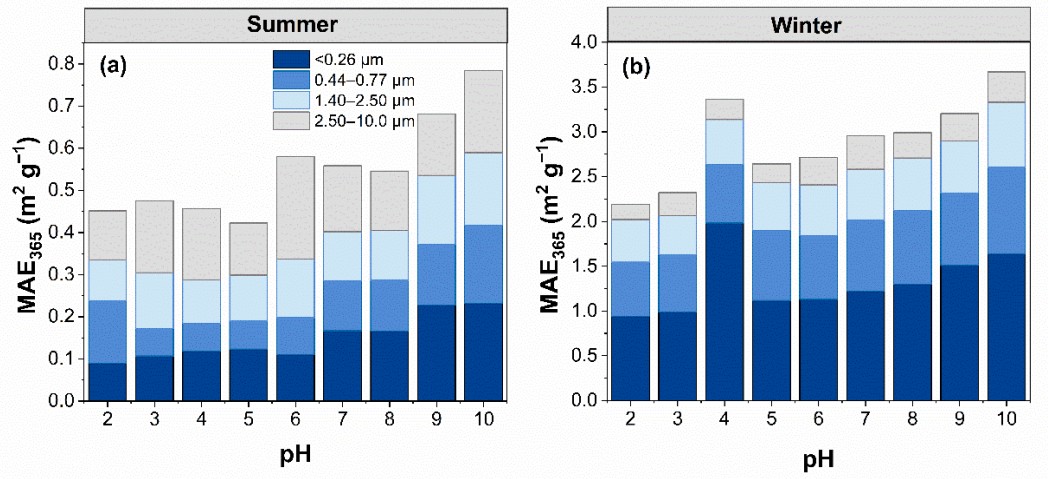

**Figure 4.** pH dependence of mass absorption efficiency ($MAE_{365}$) for WSOC in particles of different sizes in (a) summer and (b) winter in the pH range of 2–10.

### 3.2.3 Difference absorbance spectra (Δabsorbance)

Following the work of Dryer et al. (2008), the difference absorbance spectra (Δabsorbance ($\lambda$)) can identify whether specific spectral bands defined as carboxyl and phenolic groups undergo significant changes within their corresponding pH ranges. Therefore, the Δabsorbance ($\lambda$) was calculated in this study to examine the behavior of groups in resonance with chromophores in WSOC under pH titration. Based on the distribution of the $pK_a$ values of carboxyl (3.0–5.6) and phenolic groups (>8.0), carboxyl plays a major role in the absorbance in the range of approximately pH 3.0–6.0, while phenolic groups should predominate at pH>8.0.

The results summarized in Fig. 5 show that the most notable feature bands of Δabsorbance were observed at ~270 nm in almost all samples in the pH 3.0–6.0 range associated with the carboxyl group (Phillips et al., 2017), although an absorption band at ~300 nm was also observed for particle sizes of >2.50 μm in summer and winter, suggesting the presence of at least

305 two distinct chromophores deprotonated for particle sizes of >2.50 μm in this pH range (Liu et al., 2020). However, the absorption band at ~270 nm was not observed for particle sizes of 2.50–10.0 μm in summer, suggesting that the carboxyl deprotonation feature was inhibited compared to other samples. For the pH 8.0–10 range, an enhanced absorption band was primarily centered between 300 and 400 nm, and this band is approximately the position of the phenolic group, indicating that phenolic groups play a major role in this pH range (Schendorf et al., 2019).

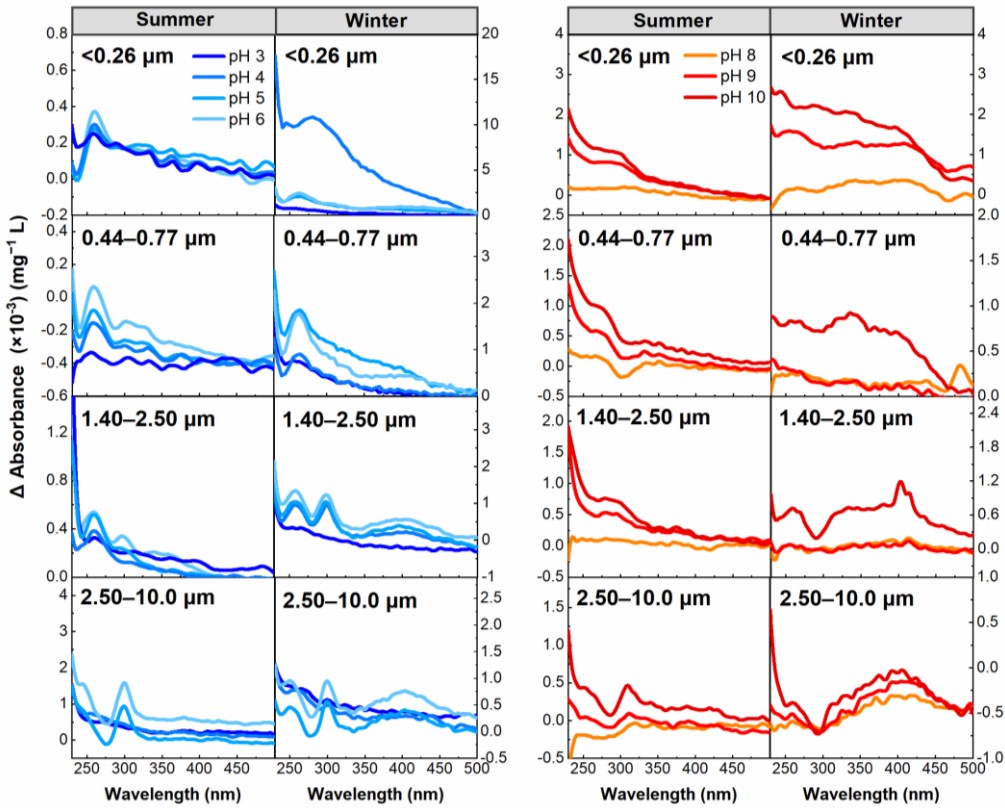

**Figure 5.** Difference absorbance spectra (Δabsorbance) of WSOC in particles of different sizes in winter and summer in the pH range of 2–10.

## 3.3 Role of pH on EEM fluorescence spectra

### 3.3.1 EEM fluorescence properties

To examine the role of pH on the fluorescence intensity and peak position of WSOC from different particle sizes, the

FI$_m$/TOC of WSOC was calculated from the EEM spectra and is plotted in Fig. 6. Overall, the FI$_m$/TOC of all samples generally decreased with increasing pH. This trend was verified by the results in Fig. 7, where from pH 2 to 10, FI/TOC first slightly increased and then significantly decreased with increasing pH. On average, the FI/TOC of < 0.26 μm, 0.44–0.77 μm, 1.40–2.50 μm, and 2.50–10.0 μm decreased by 3.8 %, 3.5 %, 4.7 %, and 6.8 %, respectively, per unit pH increase in winter, which are significantly more than those (0.6 %, 1.7 %, 0.2 %, and 2.5 %, respectively) in summer. Furthermore, a redshift of fluorescence peak positions with increasing pH was observed in summer (Fig. 6), but the opposite trend was observed in winter (blueshift). Protonation and dissociation of the acidic/basic groups in aromatic compounds can generally lead to a shift in fluorophores (Coble et al., 2014; Schulman et al., 1985). For example, the dissociation of the electron withdrawing groups (e.g., –COOH and –NO$_2$) leads to a blueshift in fluorophores, while the dissociation of the electron donating groups (e.g., –OH) results in a redshift in fluorophores (Schulman et al., 1985). In this regard, we speculate that the hydroxyl groups play a leading role in pH-responsive fluorescence in summer samples, while carboxylic and nitro groups play a dominant role in winter samples.

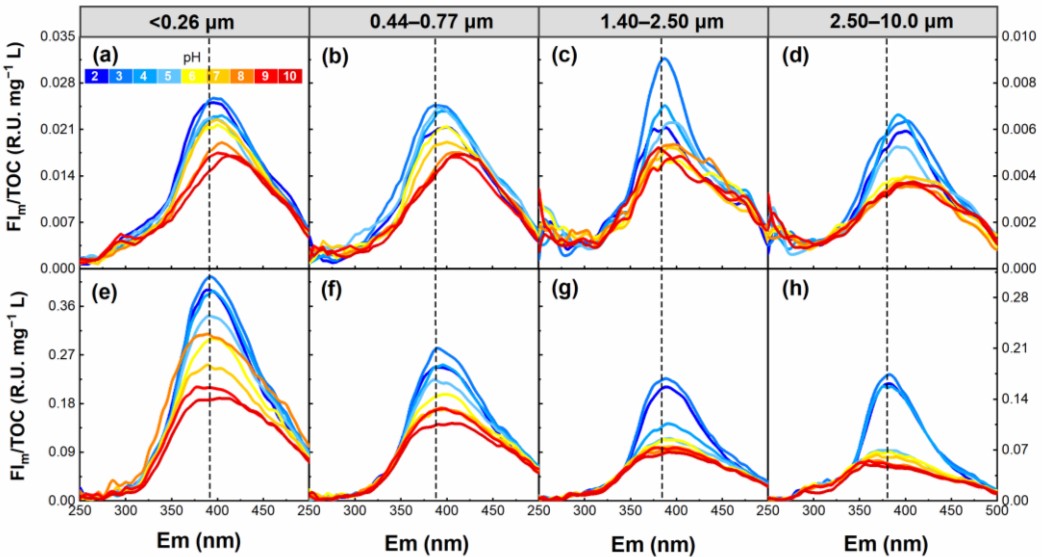

**Figure 6.** pH dependence of the FI$_m$/TOC for WSOC in particles of different sizes in summer (upper row) and winter (lower row) in the pH range of 2–10.

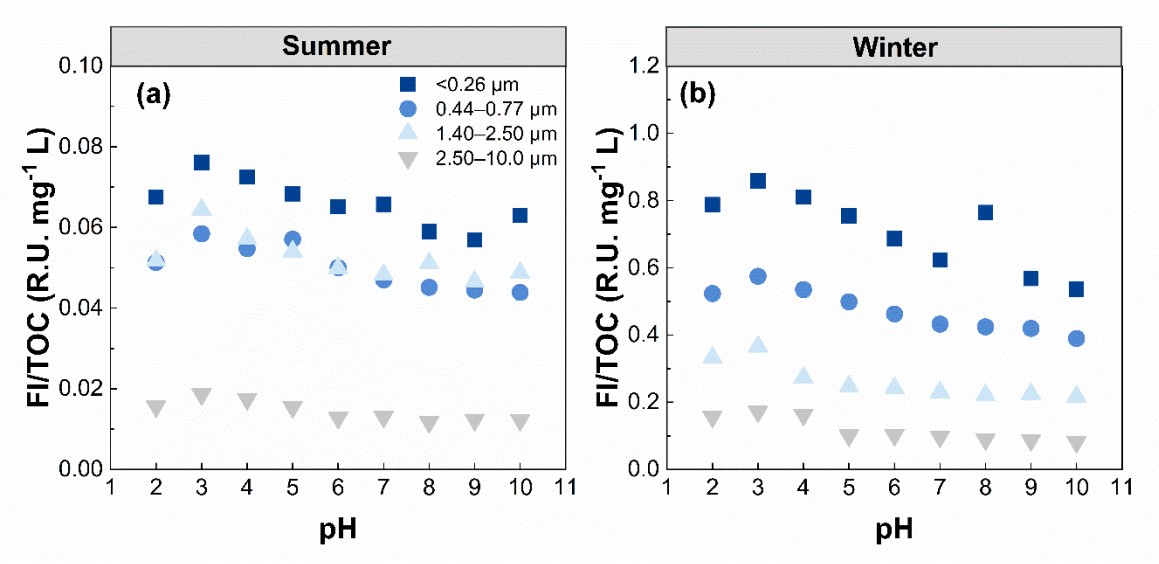

**Figure 7.** pH dependence of the FI/TOC for WSOC in particles of different sizes in (a) summer and (b) winter in the pH range of 2–10.

### 3.3.2 PARAFAC Components

To investigate the types of fluorescence components in WSOC in different particle sizes, the EEM spectra were decomposed into three fluorescent components (C1, C2, and C3), as shown in Fig. 8. The chemical components corresponding to C1 and C2 were assigned to higher oxygenated humic-like (HULIS1) and less oxygenated humic-like (HULIS2) fluorophores, respectively (Qin et al., 2018; Qin et al., 2021b; Chen et al., 2016b; Xiao et al., 2018a), while those corresponding to C3 were assigned to protein-like organic matter (Chen et al., 2016b).

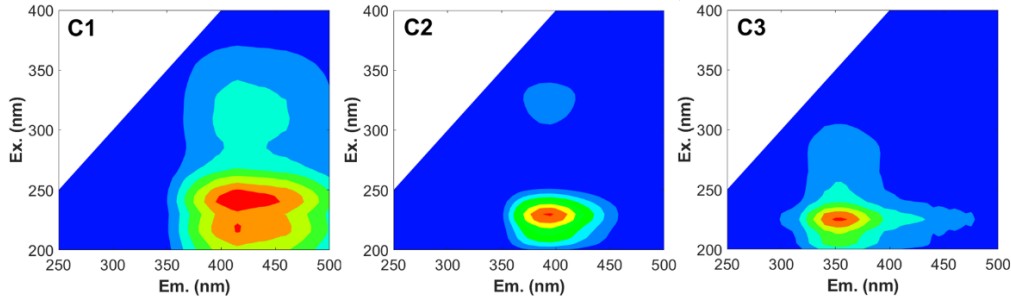

**Figure 8.** The PARAFAC analysis identified fluorescence components (C1, C2, and C3) for all WSOC samples.

The peak intensities of the fluorescent components ($F_{max}$) are shown in Fig. 9. pH also had an important effect on the $F_{max}$ of the fluorescent components of WSOC. $F_{max}$ of different fluorescent components showed a similar variation pattern with increasing pH. For example, $F_{max}$ of all fluorescence components showed a peak at pH 3 and then tended to decrease with

increasing pH. However, the magnitude of the effect of pH on $F_{max}$ of different fluorescent components was different. In contrast, HULIS2 varied significantly while HULIS1 and protein-like organic matter varied slightly with pH. The HULIS2 fluorophores showed the most susceptibility to acidity, likely indicative of a greater proportion of acidic/basic groups in HULIS2 fluorophores.

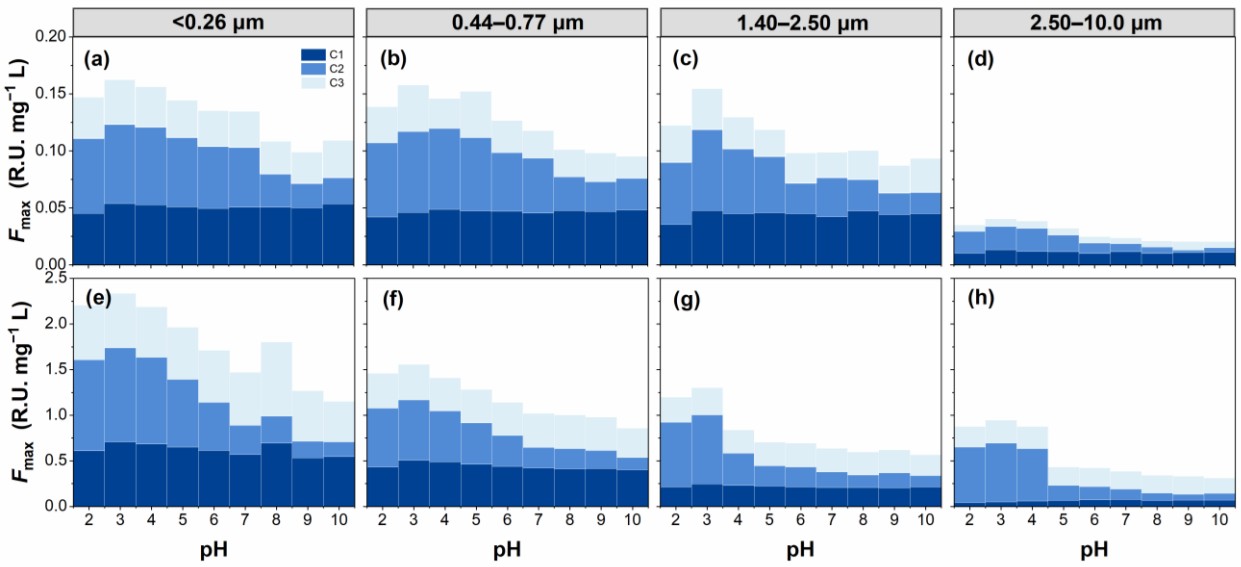

**Figure 9.** pH dependence of the $F_{max}$ of fluorescence components for WSOC in particles of different sizes in summer (upper row) and winter (lower row) in the pH range of 2–10.

### 3.3.3 Deep properties of fluorescence

AQY and Stokes shift were used to investigate the efficiency and energy change of the fluorescence process of WSOC. As shown in Fig. 10, AQY was also pH-dependent in all WSOC samples, and it generally decreased with increasing pH. This

result is similar to a conclusion reported in our previous study (Qin et al., 2021b). It has been reported that a larger rate of non-radiative transition seems to be favorable for the AQY of fluorophores (Xiao et al., 2020). Thus, our AQY data presented here

indicate the rate of non-radiative transition of the WSOC fluorophores decreased with increasing pH. Additionally, a high AQY of a substance means that only a small portion of the absorbed radiative energy will be converted into heat, thus reducing its heating effect (Aiona et al., 2018). Hence, the pH-dependent AQY presented above further confirms that the impact of WSOC on climate would be enhanced with increasing pH.

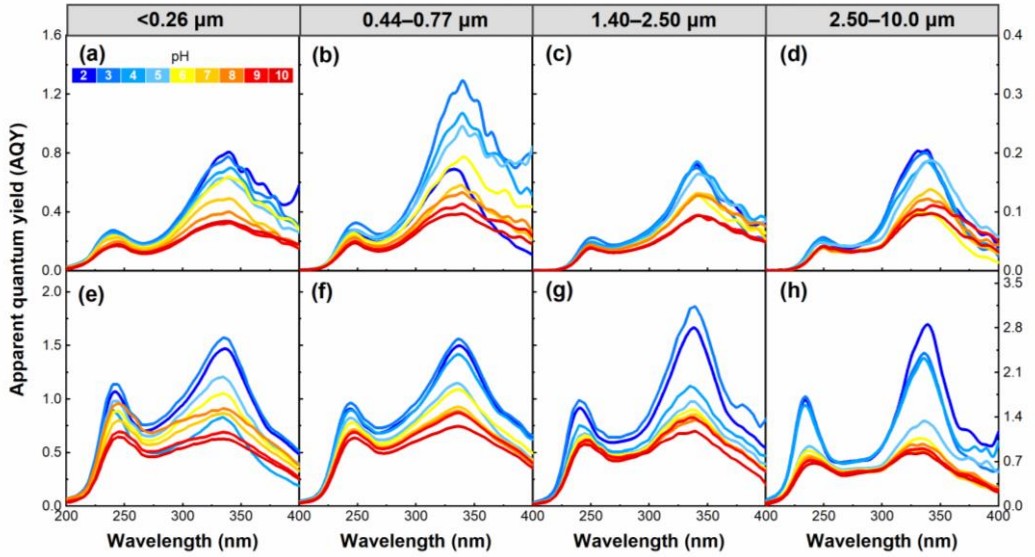

**Figure 10.** pH dependence of apparent quantum yield (AQY) for WSOC in particles of different sizes in summer (upper row) and winter (lower row) in the pH range of 2–10.

The Stokes shift is an important energy parameter of fluorescence that can be affected by the chemical environment of the fluorophore (Xiao et al., 2019). As shown in Fig. 11, there was a common feature in all samples showing WSOC having two distinct peaks at Stokes shifts around 0.7 μm and 2.0 μm. However, WSOC with lower pH tend to have higher intensity at high Stokes shift values (about 2.0 μm$^{-1}$). This trend was evident in winter samples, probably resulted from the high content of aromatic compounds in these samples (with the extensive π-conjugated system) because pH had an important impact on the π-conjugated systems and thus changed Stokes shifts of WSOC (Xiao et al., 2019). Furthermore, summer WSOC tend to have higher Stokes shifts (at about 2.0 μm$^{-1}$) at higher pH, except for particles of 2.50-10.0 μm. In contrast, winter samples usually have lower Stokes shifts (at about 2.0 μm$^{-1}$) at higher pH.

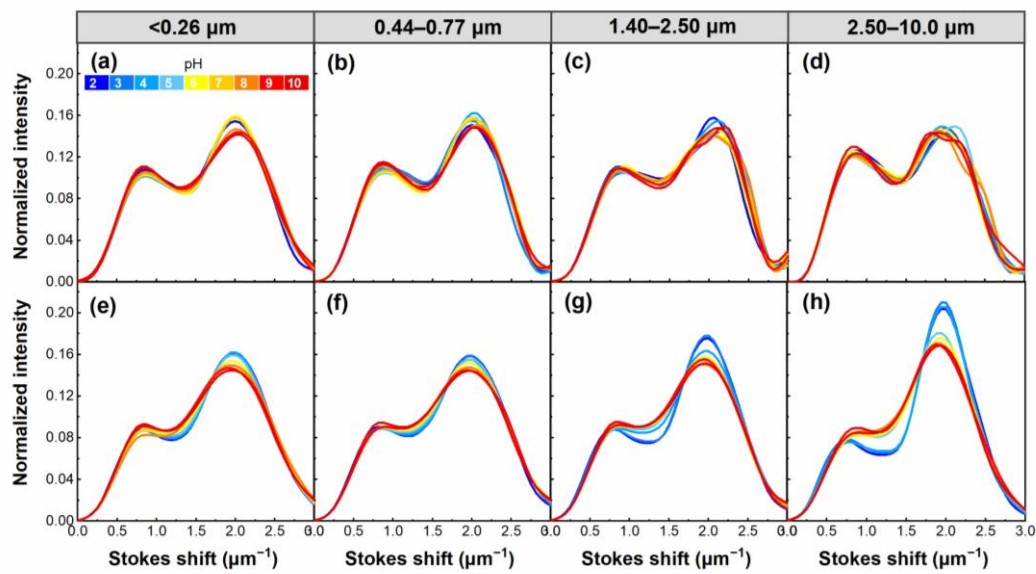

**Figure 11.** pH dependence of the Stokes shift for WSOC in particles of different sizes in summer (upper row) and winter (lower row) in the pH range of 2–10.

## 4 Summary and atmospheric implications

In this study, we examined how the chemical structures and optical properties of WSOC were affected by pH and particle size, which varied with season, source, and particle aging. For different particle sizes, higher $SUVA_{254}$, $MAE_{365}$, and FI/TOC were observed in smaller particles, suggesting the relatively higher aromaticity/molecular weight and more freshness of WSOC in smaller particles. In contrast, WSOC in larger particles underwent a series of degradation reactions during the aging process, leading to reduced aromaticity/molecular weight. The most significant seasonal differences in the optical properties (e.g., $SUVA_{254}$, $MAE_{365}$, and FI/TOC) were observed for WSOC in the smallest particles (*t*-test, $p=0.021$), suggesting significant seasonal differences in the sources and chemical structures of WSOC in small particles. WSOC in smaller particles contained more nitrogen-containing organic compounds with aromatic structures, leading to a stronger light absorption, compared to the case of larger particles. The carboxylic groups tend to be enriched in larger particles (1.40–2.50 μm), whereas the contribution of phenolic groups was the highest in smaller particles (< 0.77 μm) and the lowest in larger particles (1.40–2.50 μm), indicating that WSOC with sizes of < 0.77 μm was most likely derived from biomass burning, although the contribution from secondary

formation source cannot be completely excluded. For different pH responsive properties, $MAE_{365}$ generally increased with increasing pH, indicating WSOC had a stronger light absorption capability at high pH, in which smaller particles showed much

stronger susceptibility to pH change due to the higher content of aromatic species in their WSOC. Moreover, the FI/TOC, $F_{max}$, and AQY showed a peak at pH of 3, and then significantly decreased with increasing pH. The results of PARAFAC analysis showed that the different fluorescent components showed a similar variation pattern with increasing pH, but the HULIS2 fluorophores were most sensitive to acidity.

The results presented in this study suggest that the chemical characteristics and optical properties of WSOC with different

particle sizes can provide information on their sources and atmospheric aging processes. The variation of both $MAE_{365}$ and AQY of WSOC with increasing pH suggested the enhanced impact of WSOC on climate. Aerosol pH is often acidic and acidity decreases with increasing particle size (Battaglia et al., 2017; Craig et al., 2018). In this regard, the radiative forcing of WSOC in real atmospheric environments may be overestimated if the effect of pH is not considered, especially for WSOC in smaller particles. For example, aerosol radiative forcing may be overestimated based on an aerosol pH value of 7 instead of those at

lower values, such as pH of 2 (Pandey et al., 2020). The impact of pH on the light absorption properties is stronger for smaller particles, which may represent pH effect on fresh WSOC, as we previously reported (Qin et al., 2022). These findings have important implications in applying optical properties for identifying the chemical structures and sources of WSOC and improving the accuracy of assessing the climate effects of WSOC.

**Appendix A: Abbreviations**

| | |
|---|---|
| AAE | Absorption Ångström Exponent |
| AQY | apparent quantum yield |
| $\Delta$absorbance ($\lambda$) | difference absorbance spectra |
| BrC | brown carbon |
| EEM | three-dimensional excitation-emission matrix |
| FI/TOC | average fluorescence intensity per unit TOC |
| $FI_m$/TOC | average fluorescence intensity per unit TOC at each emission wavelength |
| $F_{max}$ | peak intensities of the fluorescent components |
| FTIR | Fourier transform infrared |

| | |
|---|---|
| HULIS | humic-like substances |
| HULIS1 | higher oxygenated humic-like |
| HULIS2 | less oxygenated humic-like |
| $MAE_{365}$ | mass absorption efficiency |
| PARAFAC | parallel factor analysis |
| RH | relative humidity |
| RU | Raman units |
| SOA | secondary organic aerosols |
| $SUVA_{254}$ | UV absorbance at a wavelength of 254 nm |
| TOC | total organic carbon |
| UV–Vis | ultraviolet–visible |
| WSOC | water-soluble organic carbon |

**Data availability**. The data used in this study are available on the Zenodo data repository platform: https://doi.org/10.5281/zenodo.7067423.

**Author contribution.** JT and YZ designed the experiments and finalized the article. YQ, JQ, and YG performed the measurements. YQ performed the data analysis and wrote the article with the assistance of KX. XW collected the samples. TQ, XZ, SS, and JL provided useful comments on the article. JG, ZZ, and RC discussed the results.

**Disclaimer.** Publisher's note: Copernicus Publications remains neutral with regard to jurisdictional claims in published maps and institutional affiliations.

**Competing interests.** The authors declare that they have no conflict of interest.

**Acknowledgments.** This work was supported by the National Natural Science Foundation of China (41675127 and 41975168).

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
