# Peer review of "Measurement Report: Investigation of pH- and particle sizedependent chemical and optical properties of water-soluble organic carbon: implications for its sources and aging processes"

_Atmospheric Chemistry and Physics, 2022_

## Editor Comment (EC1)

**Measurement report: A new coupled method…**

By Y. Qin et al.

**General comments**

In this paper, the authors report on the influence of pH and particle size on the chemical structures (i.e., functional groups, such as carboxylic, hydroxyl and phenolic) and optical properties (UV/fluorescence) of water-soluble organic carbon (WSOC) in aerosol particles, which were collected by MOUDI impactor during summer and winter. Their results provide information on WSOC sources and their aging processes in the atmosphere.

However, the content is mostly descriptive presenting the results with no deeper discussion, especially concerning atmospheric implications. In addition, the title sounds quite technical and the introduction is rather modest. Namely, there are many published articles on WSOC, also in size-segregated aerosols.

The manuscript could be of adequate atmospheric interest to merit publication in *Atmospheric Chemistry and Physics* as a *Measurement Report*, but after major revision, with addressing the following comments and/or questions.

Besides, I highly recommend the English language checking. I suggest also a separate list of abbreviations, which would be very helpful.

Data availability: Data have to be available in repository.

**Specific comments**

I suggest to change the title.

Some more references on WSOC in size-segregated aerosols should be involved in the Introduction.

Line 16: …source of these materials (not good word), source of WSOC constituents.

Line 32: "…ranging from 1 to 100 μm" (not really true). Important fraction represent particles below 1 μm. Please, check the definition.

Lines 45/46: Information on size-segregated WSOC: references are missing (e.g., Frka et al., Atmos. Environ. 2018)

Lines 74/75: Collection from June 2019 to August 2020, from 8 a.m. to 7 a.m. next day. This is one year. Can you explain how you got 82 samples only? In addition, if understood well, four size ranges have been considered (line 64), which means each sampling day 4 samples.

Line 80: Please, explain how the filters were combined for WSOC extraction.

Line 92: Why did you need to remove air from the WSOC solution?

Line 155: From the results in Table 1, I would not say that there is a big difference in $S_{275–295}$ for winter WSOC among all size ranges.

Line 156: Which results exactly show that? It is not clear.

Lines 154-172: This part is confused, and should be rewritten.
Line 158: It is written: "highlighting the relatively higher aromaticity of WSOC in small particles«;
on the other hand in line 156: "the molecular weight of WSOC was higher for particle sizes of 2.50–10.0 μm in summer and 1.40– 2.50 μm in winter",
and (line 166/167): "It has been documented that more aromaticity and larger molecular sizes of light absorbing substances tend to have higher MAE365, which is mainly derived…."

You concluded that the higher aromaticity is found in the smallest (below 0.26 μm), but on the other hand, higher molecular weight of WSOC was typical for bigger sizes.

Line 181: …"that aged WSOC undergone a growth process with increasing particle size". This statement is a bit strange. Please, correct.
Generally, with the process of growth the particle size increases.

Line 194: Both samples (which samples did you have in mind)?

Chapter 3.2.1: I suggest at least short concluding remarks from FTIR analyses.

Lines 209-210: This part can be moved in the Introduction.

Line 219: Not only phenol; so, it is better to say "aromatic compounds".

Lines 224/225: This is definitely not good enough / not a sufficient explanation. Just to remind, WSOC in winter (and summer) aerosols can be of primary and secondary source. Please, check some more references and add appropriate explanation.
For example, nitroaromatic compounds (phenolic like) can be largely formed via different reactions in gas or aqueous phase, and can be present in fine particles (below 1μm) in all seasons (See ref. e.g. Frka et al., Chemosphere 2022)

Line 232: Can you really say "monotonic"?

Line 236/237: Can you give some explanation?

Lines 243-246: Make two sentences.

Line 264: High pH values (above 7) are very rare for actual atmospheric conditions (clouds, fog, and aqueous aerosol), usually one can find more acidic conditions.

Lines 267/268: Please, give some explanation, why $MAE_{365}$ for the smallest particles changed more.

Figure 6 is not necessary here, since the results are present also in Fig.7.

Line 281: The redshift and blueshift cannot be seen.

Lines 282-285: You speculate that –COOH and –OH groups influence the most on fluorescence behavior. What about aromatic groups (line 282)?

Line 303: From Fig. 9, I would not make such a conclusion. It can be seen that for both, HULIS1 and HULIS2, $F_{max}$ decreased with pH very similarly (they had the same trend).

Lines 312-318: Not clear.

Lines 325/326: What can you say based on these results (what about aromatic compounds in winter WSOC samples)?

Chapter 4. Change the title as: "Summary and atmospheric implications"

Lines 338/339: As I said above, this is definitely not good enough. Please, correct.

Line 345: Please, see my comment above.

**Figures:** All figures/their subtitles are needed to be updated with missing information (see e.g. below for Fig.4).
Figure 1: Complete the information in the capture.
Figure 2: Complete the information in the capture.
Figure 3: Correct. In summer () and winter particles ()
Figure 4: Complete the information, for example as: Difference absorbance spectra (Δabsorbance) of WSOC in winter and summer particles of different sizes in the pH range……
Figure 5: Complete the information.
Figures 6, 7: Complete the information.

**Technical corrections**
Line 199: Correct the sentence.
Line 211:…in the range of 3.0-9.0.
Line 312: not materials/ use another word
Line 347: Delete "an indication of"

---

## Author Comment (AC1)

Dear Editor-in-Chief and Reviewers,

We are submitting a revised version of the manuscript (No.: acp-2022-321), entitled: "**Measurement Report: Investigation of pH- and particle size-dependent chemical and optical properties of water-soluble organic carbon: implications for its sources and aging processes**". We have carefully addressed all the comments provided by the reviewers. The details can be found in our enclosed "Response to Reviewers". In the attachment, an item-by-item response to the comments of the reviewers is given below. All revisions are highlighted in blue in the main text of the revised manuscript.

Thank you for taking care of the review process for this paper.

Sincerely,

Prof. Jihua Tan and coauthors

College of Resources and Environment, University of Chinese Academy of Sciences,

Beijing 100049

tanjh@ucas.ac.cn

The manuscript exhibited the effects of pH and particle size on optical properties and functional groups to identify the chemical structure, aging and sources of WSOC. Considering that the characteristics of WSOC are influenced by many factors and the complexity of atmospheric processes, this is an attractive study that can improve data support for climate effect of WSOC. There are several issues that need to be point out before publication (see additional comments):

Additional comments:

1. Line 38: Avoid lumping references as in (Yu et al., 2017; Park et al., 2015; Du et al., 2014) and all other. Instead summarize the main contribution of each referenced paper in a separate sentence. For example, "…derived from the biomass combustion and atmospheric oxidation reactions of organic compounds (Yu et al., 2017; Park et al., 2015; Du et al., 2014), …".

**Response: We are grateful for the suggestion.** We have summarized the results of each referenced in Line 40-45 and Line 51-58 according to your suggestions, and details are as follows:

Line 40-45: "WSOC is released from anthropogenic (e.g., biomass burning and coal combustion) and natural sources, and can also be formed through complex secondary reactions (Yu et al., 2017; Wu et al., 2018). The sources of WSOC vary significantly with location and season, e.g., WSOC in $PM_{2.5}$ was primarily derived from secondary aerosol formation and biomass burning in Korea (Park et al., 2015), and from coal combustion, biogenic emission, and secondary aerosol formation in winter and biogenic emission and secondary oxygenation of vehicle exhaust in summer in a northwest city of China (Qin et al., 2018)."

Line 51-58: "Therefore, the size-distribution of WSOC can serve as a good indicator of its sources, fate, and aging processes (Boreddy et al., 2021; Jang et al., 2019; Frka et al., 2018). For example, Frka et al. (2018) found that wood burning were the most important source of humic-like substances (HULIS) in the aerosol accumulation mode (from ~0.1 to ~2 μm) during the autumn and winter; Jang et al. (2019) reported that HULIS in smaller particles were likely derived from local sources while those in larger

particles were from secondary organic aerosols (SOA) in the atmosphere, and Qin et al. (2021a) found that the fluorescence properties of WSOC varied with the particle size, and the fluorescence characteristics of different particle sizes could be used to reveal the aging of WSOC."

2. Line 41-42: "…air quality and human health". Please provide references to other literature on the environmental and health effects of WSOC.

**Response: Thank you for your valuable suggestion.** We have cited related references in Line 39-40 as follows:

"Water-soluble organic carbon (WSOC, see Appendix A for a list of abbreviations) comprises a considerable fraction of organic aerosol mass (10%–80%) (Horník et al., 2021), and plays important roles on climate change (Chen et al., 2020; Sun et al., 2011) and air quality (Snyder et al., 2009)."

3. Line 75: Please supplement the information of the sampling location, such as what is the major impact on the surrounding environment, whether there is a residential area, etc.

**Response: Thank you for your above suggestion.** We have added the information of the sampling location and described them in Line 89-92 as follows:

"Particulate samples were collected on the roof of a building (~20 m above the ground) inside the campus of the University of Chinese Academy of Sciences (40°24′N, 116°40′E) in Huairou District of Beijing, China. The sampling site is in a typical urban environment surrounded by schools, research institutes, and hospitals. There were no obvious industrial sources nearby, and pollutants were mainly derived from regional transport."

4. Line 78: In Section 2.3.2, quantitative information is required to calculate SUVA, MAE and FI/TOC. Therefore, filters need to be weighed after sampling until analysis. Please supplement the weighing details.

**Response: Thank you for your comment.** Quantitative calculations of SUVA, MAE and FI/TOC can be performed without weighing the filters in this study. We are mainly focused on the relative comparison of these indices at different particle sizes in the same period, rather than absolute comparison. So, quantifying SUVA, MAE, and FI/TOC in terms of WSOC concentrations may be more appropriate, and the method of WSOC concentration analysis has been described in Line 101-104. Based on your suggestion, we will present the weighing details of filters to be included in a follow-up paper. We seek for the reviewer's tolerance and understanding. Many thanks for your kind help!

5 Line 80: "…similar to previous studies (Qin et al., 2018)". Please provide more references.

**Response: Thank you for your valuable suggestion.** Since our description here is not clear enough, we have deleted this sentence from manuscript and modified it in Line 101 as follows:

"For each particle size, a quarter of each filter of all the collected samples in summer or winter season were mixed together, extracted twice via ultrasonication in Milli-Q water for 15 min to achieve the extensive release of solubilized WSOC,"

6. Line 83: Please provide the blank concentration of WSOC in the method section.

**Response: Thanks for your kind reminder.** We have added the WSOC concentration of blank filter in Line 104 as follows:

"The average WSOC concentration of the blank filter was 0.39 mg $L^{-1}$."

7. Line 85 and Line 95: "HCI" should be "HCl".

**Response: We are grateful for the suggestion.** We have replaced all "HCI" with "HCl" in the revised manuscript as follows:

Line 107: "HCl and NaOH were used to adjust the pH of the WSOC solutions."

Line 119: "a drop of 2 mol $L^{-1}$ HCl was added to the WSOC solution to remove the interference of inorganic carbon."

8. Line 88: Section 2.1.3: Is the pH characteristic of the sample lost after adjusting the pH value of WSOC, then it can not represent the environmental samples? Can we use the pH value of the sample?

**Response: Thank you for your careful review.** The point you mentioned is really important. Previous study has demonstrated that the change in fluorescence from pH 2-12 are reversible (Henderson et al., 2009). Therefore, the pH fluorescence properties of WSOC would not disappear after adjusting the pH of WSOC.

The pH of particulate matter from different sources varies widely (0-9) (Fridlind and Jacobson., 2000; Qin et al., 2021), and the pH of the same sample also varies significantly during atmospheric chemistry processes (Caig et al., 2018). In these cases, a pH range of 2-10 was selected for our study in order to comprehensively investigate the effect of pH on the fluorescence properties of WSOC. Furthermore, the effect of pH on the fluorescence properties of WSOC with different particle sizes has not been reported yet. The chemical components and structures of different particle size samples were different. Using the pH of the samples for the study may lead to difficulties in interpreting the mechanism of the effect of pH on the fluorescence properties of WSOC. But you are also right in your consideration, we will consider researching the effect of raw pH of sample on the fluorescence properties of different particle sizes in further study according to your suggestions.

Reference:

1. Henderson R K, Baker A, Murphy K R, et al. Fluorescence as a potential monitoring tool for recycled water systems: A review[J]. Water Res, 2009, 43(4): 863-881.

2. Fridlind, A. M.; Jacobson, M. Z. A study of gas-aerosol equilibrium and aerosol pH in the remote marine boundary layer during the first aerosol characterization experiment (ACE 1). J. Geophys. Res. 2000, 105(D13), 17325-17340.

3. Qin, Y., Yang, Y., Qin, J, et al. pH-responsive fluorescence EEM to titrate the interaction between fluorophores and acid/base groups in water-soluble organic compounds of $PM_{2.5}$. Environ. Sci. Technol. Lett. 2021, 8(2), 108-113.

4. Caig R L, Peterson P K, Lucy N, et al. Direct determination of aerosol pH: size-resolved measurements of submicrometer and supermicrometer aqueous particles[J]. Anal. Chem. 2018, 90(19), 11232–11239.

9. Line 89: "50 ml" should be "mL".

**Response: We are extremely grateful for pointing out this problem.** We have revised it in Line 113 as follows:

"a 50 mL WSOC solution was first acidified to a pH below 3 by addition of HCl solution,"

10. Line 100: Section 2.2.2, these quality assurance and control (QA/QC) procedures for these experiments should be explained in details.

**Response: Thanks for your kind reminder.** We have added quality assurance and control procedures for these experiments in Section 2.4 according to your suggestion as follows:

"**2.4. Quality assurance and quality control**

Quality assurance and quality control (QA/QC) procedures were applied through the laboratory and instrumental analysis processes. Before sampling, all quartz fiber filters were baked at 550°C for 5.5 h. Prior to analysis, all vials were washed with Milli-Q water, dried, and preheated at 550°C for 5.5 h in a muffle furnace, and then wrapped in aluminum foil before being used. WSOC extraction and measurement were conducted in a designated laboratory. All WSOC concentrations, UV–Vis absorption spectra, EEM fluorescence spectra, and functional groups data were blank-corrected. All instruments operated in this study require regular calibration. Sterile gloves and robes were worn throughout the experimental procedure to avoid contamination and to ensure the accuracy of the test results."

11. Line 130: In Section 2.3.3, please provide how Raman scattering and Rayleigh scattering are removed in EEM.

**Response: Thank you very much for your valuable advice.** In this work, we used an interpolation technique to eliminate Rayleigh and Raman scatterings from EEM fluorescence spectra. We have described it in Line 154-155 as follows:

"The interfering fluorescence signals of the Rayleigh and Raman scattering were then eliminated by an interpolation technique."

12. Line 166-170: The author showed the difference of WSOC sources through different MAE, so please give a specific value or range in order to more intuitively express the MAE difference between biomass burning and secondary formation. In addition, the authors has only discussed biomass burning and secondary formation before, however, it points out three primary sources (biomass burning, coal combustion, and vehicle exhaust) at the end of this sentence (Line 172), please elaborate on this point.

**Response: Thank you for your comment.** The range of $MAE_{365}$ values for WSOC from different sources have been provided in previous studies (Geng et al., 2020; Washenfelder et al., 2015), the $MAE_{365}$ of WSOC from biomass burning was 0.76-2.47 $m^2\ g^{-1}$, 0.20-1.33 $m^2\ g^{-1}$ for fossil fuel combustion, and -0.11±0.20 $m^2\ g^{-1}$ for biogenic SOA. Following your suggestion, we have added the range to demonstrate the difference in $MAE_{365}$ between biomass burning and secondary formation in in Line 215-216.

The $MAE_{365}$ values of < 0.26 μm, 0.44–0.77 μm, 1.40–2.50 μm, and 2.50–10.0 μm were 0.1258, 0.1321, 0.1014, and 0.1145 $m^2\ g^{-1}$ in summer and 1.2615, 0.7991, 0.8206, and 0.3707 $m^2\ g^{-1}$ in winter. Previous study has found that the $MAE_{365}$ of WSOC from biomass burning was 0.76-2.47 $m^2\ g^{-1}$, 0.20-1.33 $m^2\ g^{-1}$ for fossil fuel combustion, and -0.11±0.20 $m^2\ g^{-1}$ for biogenic SOA (Geng et al., 2020). Therefore, we speculate that WSOC may be mainly derived from secondary formation in summer and from mixed sources of biomass burning and fossil fuel combustion in winter. According to your suggestion, we have carefully revised it in Line 215-219 and now described as:

"Literature reported $MAE_{365}$ from different sources showed values of 0.76–2.47 $m^2\ g^{-1}$ for biomass burning smoke, 0.20–1.33 $m^2\ g^{-1}$ for fossil fuel combustion, and -0.11±0.20

$m^2$ $g^{-1}$ for biogenic SOA (Geng et al., 2020). Our previous study also found higher $MAE_{365}$ from combustion sources than ambient samples (Qin et al., 2022). The $MAE_{365}$ values observed in the present study suggest that WSOC may be mainly derived from secondary formation in summer and from mixed sources of primary emissions (e.g., biomass burning and fossil fuel combustion) in winter."

Reference:

1. Geng X, Mo Y, Li J, et al. Source apportionment of water-soluble brown carbon in aerosols over the northern South China Sea: Influence from land outflow, SOA formation and marine emission[J]. Atmospheric Environ. 2020, 229.117484.

2. Washenfelder R. A., Attwood A. R., Brock. C. A, et al. Biomass burning dominates brown carbon absorption in the rural Southeastern United States[J]. Geophys. Res. Lett. 2015, 42, 653–664.

13. Line 175: The unit of "Stokes shift" –"um$^{-1}$" should be "$\mu m^{-1}$".

**Response: Sorry for our mistake.** The corresponding revision has been presented in Table 1 as follows:

"Stokes shift ($\mu m^{-1}$)"

14. Line 186, Line 340 etc: The article has repeatedlly mentioned statistical terms, such as "significant" or "significantly". Therefore, when make the comparison of data between seasons, authors should supplement the statistical analysis in Section 2(Experimental methods) and provide the statistical results (t-test or ANOVA) in Section 3(Results and discussion).

**Response: Thank you for your comment.** As suggested by the reviewer, we employed independent samples $t$-tests to evaluate seasonal differences of the data. Descriptions of the statistical analysis for data are shown in Line 172. The results of the $t$-tests are presented in Line 223 and Line 390 of the revised manuscript as follows:

Line 172: "Additionally, the independent-samples $t$-test was used to evaluate the seasonal differences in light absorption and fluorescence properties of WSOC."

Line 223: "Furthermore, there are significant seasonal variations in the optical properties (e.g., SUVA$_{254}$, MAE$_{365}$, and FI/TOC) of WSOC in smaller particles (*t*-test, *p*=0.021),"

Line 390: "The most significant seasonal differences in the optical properties (e.g., SUVA$_{254}$, MAE$_{365}$, and FI/TOC) were observed for WSOC in smallest particles (*t*-test, *p*=0.021),"

15. Line 199-200: Please supplement references.

**Response: Thank you for your valuable suggestion.** We have cited the related reference in Line 234 as follows:

"A peak at 1641 cm$^{-1}$ was also previously reported, which was attributed to conjugated carbonyl (C=O) groups and aromatic rings (C=C) (Zhang et al., 2022)."

Reference:

Zhang T, Huang S, Wang D, et al. Seasonal and diurnal variation of PM$_{2.5}$ HULIS over Xi'an in Northwest China: Optical properties, chemical functional group, and relationship with reactive oxygen species (ROS)[J]. Atmos. Environ, 2022, 268: 118782.

16. Line 205: The title showed the identificantion of structure by pH value and paticle sizes, but author only reflects the influence of particle sizes on functional groups in Section 3.1.2. Please provide the pH value of Figure 1. Additionally, please provide the effect of different pH values on functional groups (Figure and text description).

**Response: Thank you for the above suggestion.** Indeed, we have discussed the effect of different pH on the functional groups via the variation of EEM fluorescence and UV–Vis absorption spectra properties in Sections 3.2 and 3.3. The variation of UV–Vis absorption spectra with pH showed the presence of at least two distinct chromophores (carboxyl and phenolic groups) deprotonated for particle sizes of >2.50 μm in pH 3-6, while phenolic groups should predominate at pH>8.0. The change of EEM fluorescence revealed that the hydroxyl groups play a leading role in pH-responsive fluorescence in

summer samples, while carboxylic and nitro groups play a dominant role in winter samples. It will be more profound if we get the effect of pH on FTIR spectra results, as you said. We previous investigated the effect of pH on the functional groups of WSOC in $PM_{2.5}$ using FTIR spectroscopy (Wang et al., 2021). However, the determination of FTIR spectra of different pH requires a large mass of samples. Therefore, after considering the large sample requirement or other parameters, the effect of pH on FTIR spectral analysis weren't conducted in this study. The referee's concern is of importance for our further study.

We have added the pH of Figure 1 in Section 2.1.3 (Line 109-111) and modified the title of Figure 1 in Line 246 as follows:

Line 109-111: "The raw pH of WSOC with particle sizes of < 0.26 μm, 0.44–0.77 μm, 1.40–2.50 μm, and 2.50–10.0 μm was 5.61, 5.75, 5.89, and 6.19, respectively, in summer, and 6.19, 6.49, 6.64, and 6.83, respectively, in winter."

Line 246: "**Figure 1.** FTIR spectra of WSOC in particles of different sizes in summer and winter at raw pH."

Reference:

Wang X, Qin Y, Qin J, et al. Spectroscopic insight into the pH-dependent interactions between atmospheric heavy metals (Cu and Zn) and water-soluble organic compounds in $PM_{2.5}$[J]. Science of The Total Environment, 2021, 767:145261.

17. Line 209-210: "pH titration enables qualitative and quantitative analyses of functional groups on the surface of substances (Zhang et al., 2011; Xiao et al., 2014)." Suggest put it in Section 2 (Experimental methods). "However, measurements of such type have not yet been performed for particles with different sizes." Suggest put it in Section 1(Introduction) and supplement it appropriately.

**Response: We are appreciative of the reviewer's suggestion.** According to your suggestion, we have moved the sentence "However, measurements of such type have not yet been performed for particles with different sizes." to Line 78 of Section 1 (Introduction), and have revised and supplemented it carefully. Additionally, Reviewer

**2 also pointed out this problem. However, they suggested "pH titration enables qualitative and quantitative analyses of functional groups on the surface of substances (Zhang et al., 2011; Xiao et al., 2014)." is better placed in Section 1 (Introduction). After careful consideration, we have made the following modifications in Line 69-78. Thank you for input on these changes.**

"The commonly used analytical methods to characterize the optical properties of WSOC are three-dimensional excitation-emission matrix (EEM) spectroscopy and ultraviolet–visible (UV–Vis) absorption spectroscopy (Zhang et al., 2021; Yang et al., 2020). The EEM spectroscopy is a rapid as well as informative method to identify chromophores that may not be distinguished by UV–Vis absorption spectroscopy (Chen et al., 2019; Xiao et al., 2020). Therefore, EEM spectroscopy has been widely applied in atmospheric WSOC characterization (Fu et al., 2015; Qin et al., 2018). However, such a technique has not been widely applied to investigate the fluorescence properties of WSOC in different particle sizes. Fourier transform infrared (FTIR) spectroscopy has been frequently used for the identification of WSOC functional groups (Chen et al., 2016a), although this analysis is difficult to perform quantitatively. pH titration enables qualitative and quantitative analyses of functional groups on the surface of substances (Zhang et al., 2011; Xiao et al., 2014), and this approach has recently been successfully applied to the characterization of WSOC in ambient $PM_{2.5}$ (Qin et al., 2021b), but not yet on size-resolved WSOCs."

18. Line 223-255: Although phenols account for a high proportion in winter, it seems that the proportion of carboxylic acids cannot be ignored in the particle size range of 1.40-2.50 μm. In addition, although the proportion of carboxylic acid groups in summer is higher than that in winter, the proportion of phenolic groups is greater than 50% in both winter and summer, except for 1.40-2.50 μm in winter. The explanation in Line 223-225 is insufficient, and it is suggested to revise this sentence.

**Response: Thank you for your careful review.** We apologize for not describing it clearly. We have added the description of the proportion of carboxylic and phenolic groups in Line 261-264 as follows:

"Overall, the contribution of strong phenolic groups (>54%) was higher than that of carboxyl groups in all samples except particle sizes of 1.40–2.50 μm. This result indicates that phenolic groups were more abundant than carboxylic groups in WSOC. Meanwhile, except for particle sizes of 1.40–2.50 μm, the contribution of carboxyl was higher and the phenolic group was lower in summer than winter."

19. Line 281: A redshift of fluorescence peak positions with increasing pH can be observed in summer. The phenomenon could not be seen in Figure 7. Is it Figure 6?

**Response: Sorry for causing such confusion.** As you said, a redshift of fluorescence peak with increasing pH for the summer samples can be observed in Figure 6, rather than in Figure 7. We have corrected it in Line 324 as follows:

"a redshift of fluorescence peak positions with increasing pH was observed in summer (Fig. 6),"

20. Line 290: It is advised to choose the unified color represents each size. For example, Figure 5 and Figure 7.

**Response: Thanks for your kind reminder.** We have re-plotted Figure 4 and Figure 7 to make the color representing the different particle size samples uniform, and details are as follows:

[Figure]

**Figure 4.** pH dependence of mass absorption efficiency (MAE$_{365}$) for WSOC in particles of different sizes in (a) summer and (b) winter in the pH range of 2–10.

[Figure]

**Figure 7.** pH dependence of the FI/TOC for WSOC in particles of different sizes in (a) summer and (b) winter in the pH range of 2–10.

21. Line 312: Please provide references.

**Response: Thanks for your kind reminder.** We have revised it and cited related references in Line 365 as follows:

"Additionally, a high AQY of a substance means that only a small portion of the absorbed radiative energy will be converted into heat, thus reducing its heating effect (Aiona et al., 2018)."

22. Line 371: Some newly references should be cited in this paper, such as: Light absorption properties and molecular profiles of HULIS in PM$_{2.5}$ emitted from biomass burning in traditional "Heated Kang" in Northwest China. Science of the Total Environment, 2021, 776, 146014-146022. Seasonal and diurnal variation of PM2.5 HULIS over Xi'an in Northwest China: Optical properties, chemical functional group, and relationship with reactive oxygen species (ROS). Atmospheric Environment 268. Optical properties, chemical functional group, and oxidative activity of different polarity levels of water-soluble organic matter in PM$_{2.5}$ from biomass and coal combustion in rural areas in Northwest China. Atmospheric Environment, 283, (2022)119179

**Response: Thanks for your reminder.** These references have been included in the revised manuscript. Specific references are listed as follows:

Line 70: "The commonly used analytical methods to characterize the optical properties of WSOC are three-dimensional excitation-emission matrix (EEM) spectroscopy and ultraviolet–visible (UV–Vis) absorption spectroscopy (Zhang et al., 2021; Yang et al., 2020)."

Line 192: "The average $MAE_{365}$ values of particles in the size range of <0.26 μm, 0.44–0.77 μm, 1.40–2.50 μm, and 2.50–10.0 μm were 0.6937, 0.4656, 0.4610, 0.2426 $m^2\,g^{-1}$, respectively, indicating that WSOC in smaller particles had stronger light absorption capabilities (Huang et al., 2022)."

Line 234: "A peak at 1641 $cm^{-1}$ was also previously reported, which was attributed to conjugated carbonyl (C=O) groups and aromatic rings (C=C) (Zhang et al., 2022)."

Reference:

1. Zhang T, Shen Z, Zeng Y, et al. Light absorption properties and molecular profiles of HULIS in $PM_{2.5}$ emitted from biomass burning in traditional "Heated Kang" in Northwest China[J]. Science of The Total Environment. 2021, 776: 146014.

2. Huang S, Luo Y, Wang X, et al. Optical properties, chemical functional group, and oxidative activity of different polarity levels of water-soluble organic matter in $PM_{2.5}$ from biomass and coal combustion in rural areas in Northwest China[J]. Atmospheric Environment, 2022, 283: 119179.

3. Zhang T, Huang S, Wang D, et al. Seasonal and diurnal variation of $PM_{2.5}$ HULIS over Xi'an in Northwest China: Optical properties, chemical functional group, and relationship with reactive oxygen species (ROS)[J]. Atmospheric Environment, 2022, 268: 118782.

23. References: Please check the references and unified format. For example, "Effect of the Urban Heat Island on Aerosol pH" in Line 378.

**Response: We are very sorry for our careless mistake.** We have checked all references and revised the format to make them uniform. Some of the references that were modified are as follows:

"Battaglia, M. A., Douglas, S., and Hennigan, C. J.: Effect of the urban heat island on

aerosol pH, Environ. Sci. Technol., 51, 13095–13103, http://dx.doi.org/10.1021/acs.est.7b02786, 2017

Snyder, D. C., Rutter, A. P., Collins, R., Worley, C., and Schauer, J. J.: Insights into the origin of water soluble organic carbon in atmospheric fine particulate matter, Aerosol Sci. Technol., 43, 1099–1107, http://dx.doi.org/10.1080/02786820903188701, 2009."

---

## Author Comment (AC2)

Dear Editor-in-Chief,

We are submitting a revised version of the manuscript (No.: acp-2022-321), entitled: "**Measurement Report: Investigation of pH- and particle size-dependent chemical and optical properties of water-soluble organic carbon: implications for its sources and aging processes**". We have carefully addressed all the comments provided by the editor. The details can be found in our enclosed "Response to editor". In the attachment, an item-by-item response to the comments of the editor is given below. All revisions are highlighted in blue color in the main text of the revised manuscript.

Thank you for taking care of the review process for this paper.

Sincerely,

Prof. Jihua Tan and coauthors

College of Resources and Environment, University of Chinese Academy of Sciences, Beijing 100049

tanjh@ucas.ac.cn

**Response to Editor:**

**We appreciate the editor and reviewer for his positive and constructive comments and suggestions on our manuscript.** We have addressed all the comments carefully as detailed below. The grammars of whole article and some other mistakes were checked and modified simultaneously. A list of abbreviations has also been added in Appendix A. We have uploaded the raw data in the repository. The original comments are in black and our replies in blue, we also put the revised paragraph after each reply to show the changes.

Editor

General comments

In this paper, the authors report on the influence of pH and particle size on the chemical structures (i.e., functional groups, such as carboxylic, hydroxyl and phenolic) and optical properties (UV/fluorescence) of water-soluble organic carbon (WSOC) in aerosol particles, which were collected by MOUDI impactor during summer and winter. Their results provide information on WSOC sources and their aging processes in the atmosphere.

However, the content is mostly descriptive presenting the results with no deeper discussion, especially concerning atmospheric implications. In addition, the title sounds quite technical and the introduction is rather modest. Namely, there are many published articles on WSOC, also in size-segregated aerosols.

The manuscript could be of adequate atmospheric interest to merit publication in Atmospheric Chemistry and Physics as a Measurement Report, but after major revision, with addressing the following comments and/or questions.

Besides, I highly recommend the English language checking. I suggest also a separate list of abbreviations, which would be very helpful.

Data availability: Data have to be available in repository.

Specific comments

1. I suggest to change the title.

**Response: We are grateful for the suggestion.** According to your suggestion, we have changed the current title "Measurement Report: A new coupled method of pH titration and size-resolved analysis to identify the structure, aging, and source of water-soluble organic carbon" to "Measurement Report: Investigation of pH- and particle size-dependent chemical and optical properties of water-soluble organic carbon: implications for its sources and aging processes" in Line 1.

2. Some more references on WSOC in size-segregated aerosols should be involved in the Introduction.

**Response: Thank you very much for your valuable advice.** We have added more references on WSOC in size-segregated aerosols in the Introduction and revised this part appropriately in Line 46-60 as follows:

"The formation and transformation of WSOC in atmospheric particulate matter is highly complex. Existing studies mostly focused on investigating the concentration levels and optical and chemical characteristics of bulk WSOC in $PM_{2.5}$ (Xiang et al., 2017; Ma et al., 2022). Studies focusing on size-resolved WSOC are still limited, e.g., earlier studies focused on exploring the size distribution of WSOC (Timonen et al., 2008; Deshmukh et al., 2016), and a few recent studies investigated the optical properties of size-resolved WSOC (Chen et al., 2019; Qin et al., 2021a). The sources, formation mechanisms, and transformation processes of WSOC are strongly related to particle size distribution (Chen et al., 2019). Therefore, the size-distribution of WSOC can serve as a good indicator of its sources, fate, and aging processes (Boreddy et al., 2021; Jang et al., 2019; Frka et al., 2018). For example, Frka et al. (2018) found that wood burning were the most important source of humic-like substances (HULIS) in the aerosol accumulation mode (from ~0.1 to ~2 μm) during the autumn and winter; Jang et al. (2019) reported that HULIS in smaller particles were likely derived from local sources while those in larger particles were from secondary organic aerosols (SOA) in the atmosphere, and Qin et al. (2021a) found that the fluorescence properties of WSOC varied with the particle size, and the fluorescence characteristics of different particle sizes could be used to reveal the aging of WSOC. To date, a knowledge gap still remains

regarding the aging, fate, and important atmospheric processes modulating chemical and physical properties of WSOC due to the dearth of WSOC-focused research in different particle sizes and the limitations of analytical methods."

3. Line 16: ...source of these materials (not good word), source of WSOC constituents.

**Response: We are extremely grateful for pointing out this problem.** We have replaced "these materials" with "WSOC constituents" in Line 18 as follows:

"This study investigates the coupled effects of pH and particle size on the chemical structures (functional groups) and optical properties (UV/fluorescence properties) of WSOC and to further explore the structure, aging, and source of WSOC constituents."

4. Line 32: "...ranging from 1 to 100 μm" (not really true). Important fraction represent particles below 1 μm. Please, check the definition.

**Response: Thank you for pointing out this mistake.** You are right that atmospheric particulate matter includes particle smaller than 1 μm in size. We have thoroughly rewritten this paragraph in Line 38-45 and removed this sentence from manuscript as follows:

"Water-soluble organic carbon (WSOC, see Appendix A for a list of abbreviations) comprises a considerable fraction of organic aerosol mass (10%–80%) (Horník et al., 2021), and plays important roles on climate change (Chen et al., 2020; Sun et al., 2011) and air quality (Snyder et al., 2009). WSOC is released from anthropogenic (e.g., biomass burning and coal combustion) and natural sources, and can also be formed through complex secondary reactions (Yu et al., 2017; Wu et al., 2018). The sources of WSOC vary significantly with location and season, e.g., WSOC in $PM_{2.5}$ was primarily derived from secondary aerosol formation and biomass burning in Korea (Park et al., 2015), and from coal combustion, biogenic emission, and secondary aerosol formation in winter and biogenic emission and secondary oxygenation of vehicle exhaust in summer in a northwest city of China (Qin et al., 2018)."

5. Lines 45/46: Information on size-segregated WSOC: references are missing (e.g.,

Frka et al., Atmos. Environ. 2018).

**Response: Thanks for your reminder.** More references have been included in the revised manuscript in Line 51-54 as follows:

"Therefore, the size-distribution of WSOC can serve as a good indicator of its sources, fate, and aging processes (Boreddy et al., 2021; Jang et al., 2019; Frka et al., 2018). For example, Frka et al. (2018) found that wood burning were the most important source of humic-like substances (HULIS) in the aerosol accumulation mode (from ~0.1 to ~2 μm) during the autumn and winter;"

Reference:

Frka, S., Grgić, I., Turšič, J., Gini, M. I., and Eleftheriadis, K.: Seasonal variability of carbon in humic-like matter of ambient size-segregated water soluble organic aerosols from urban background environment, Atmos. Environ., 173, 239-247, http://dx.doi.org/10.1016/j.atmosenv.2017.11.013, 2018.

6. Lines 74/75: Collection from June 2019 to August 2020, from 8 a.m. to 7 a.m. next day. This is one year. Can you explain how you got 82 samples only? In addition, if understood well, four size ranges have been considered (line 64), which means each sampling day 4 samples.

**Response: Sorry for our mistakes.** Actually, we collected a total of 82 sets of six-stage size-segregated aerosol samples during the months of June 2019 to August 2020, instead of only 82 samples were collected. Among them, 46 sets of six-stage size-segregated aerosol samples during two summer and one winter months from June 2019 to August 2020. To collect sufficient mass of particulate matter for the experiment, sampling duration for samples was selected based on the degree of air pollution with a minimum of 1 day on polluted days to a maximum of 7 days on clean days, but all the sampling started at from 8:00 a.m. and ended at 7:00 a.m in 1-7 days. We apologize for our carelessness. We have made the following modifications in Line 94-99 according to your suggestion:

"A total of 46 sets of six-stage size-segregated aerosol samples were collected during

two summer and one winter months from June 2019 to August 2020, with 35 sets of six-stage size-segregated aerosol samples in summer and 11 sets of six-stage size-segregated aerosol samples in winter. To collect sufficient mass of particulate matter in each sample, sampling duration for each individual sample was selected based on the degree of air pollution with a minimum of 1 day on polluted days to a maximum of 7 days on clean days, but all the sampling started at from 8:00 a.m. and ended at 7:00 a.m in 1-7 days."

7. Line 80: Please, explain how the filters were combined for WSOC extraction.

**Response: Thanks for your kind reminder.** For each particle size sample, combined extraction of WSOC by mixing a quarter of each filter of all collected samples in summer or winter season in a vial, and then ultrasonicated for 15 min in Milli-Q water. We have made the following modifications in Line 101 and now described as:

"For each particle size, a quarter of each filter of all the collected samples in summer or winter season were mixed together, extracted twice via ultrasonication in Milli-Q water for 15 min to achieve the extensive release of solubilized WSOC,"

8. Line 92: Why did you need to remove air from the WSOC solution?

**Response: Thank you very much for your question.** The WSOC solution was bubbled with pure $N_2$ to remove $CO_2$ from air to eliminate the effect of $CO_2$ on the pH titration. We have made the following modifications in Line 115 according to your suggestion:

"Throughout the titration, the WSOC solution was bubbled with pure $N_2$ to remove $CO_2$ from air."

9. Line 155: From the results in Table 1, I would not say that there is a big difference in $S_{275-295}$ for winter WSOC among all size ranges.

**Response: Thank you for your careful review.** Previous study found that the $S_{275-295}$ of the high-molecular-weight dissolved organic matter (DOM) were generally lower than those for the low-molecular-weight (Helms et al., 2008). Therefore, we used $S_{275-}$

$_{295}$ to characterize the molecular weight of WSOC. However, we neglected that the origins and formation mechanisms of aquatic DOM and aerosol WSOC are different, which implies that examining the chemical characteristics of aerosol WSOC based on our knowledge of $S_{275-295}$ values of aquatic DOM might be speculative. Therefore, after careful consideration, we have removed $S_{275-295}$ values from Table 1 and thoroughly rewritten this paragraph in Line 185-196, which is now described as:

"The light absorption and fluorescence properties of WSOC varied with particle size (Table 1). Higher $SUVA_{254}$ for particles of < 0.26 μm than other particle sizes highlighted the relatively higher aromaticity/molecular weight of WSOC in smaller particles (Cawley et al., 2013). Baduel et al. (2011) found that HULIS undergoes degradation under UV and ozone conditions, which in turn reduces its aromaticity and molecular weight. Thus, the higher $SUVA_{254}$ values in smaller particle in this study may be due to fresh WSOC in these particle size ranges. In contrast, WSOC in larger particles undergoes a series of degradation reactions during the aging process, resulting in a decrease in aromaticity/molecular weight. The average $MAE_{365}$ values of particles in the size range of <0.26 μm, 0.44–0.77 μm, 1.40–2.50 μm, and 2.50–10.0 μm were 0.6937, 0.4656, 0.4610, 0.2426 $m^2\ g^{-1}$, respectively, indicating that WSOC in smaller particles had stronger light absorption capabilities (Huang et al., 2022). This is because WSOC in smaller particles contained more chromophores, such as nitrogenous chromophores (see Section 3.1.2), resulting in stronger light absorption capabilities (Wu et al., 2018). The average AAE values were highest in particle sizes of <0.26 μm, suggesting the existence of more wavelength dependence light absorption properties in smaller particles (Wu et al., 2018)."

Reference:

1. Helms J R, Stubbins A, Ritchie J D, et al. Absorption spectral slopes and slope ratios as indicators of molecular weight, source, and photobleaching of chromophoric dissolved organic matter[J]. Limnology and Oceanography, 2008, 53(3): 955-969.

10. Line 156: Which results exactly show that? It is not clear.

**Response: Sorry for making such confusion.** We have carefully revised this part in Line 185-196 as follows:

"The light absorption and fluorescence properties of WSOC varied with particle size (Table 1). Higher $SUVA_{254}$ for particles of < 0.26 μm than other particle sizes highlighted the relatively higher aromaticity/molecular weight of WSOC in smaller particles (Cawley et al., 2013). Baduel et al. (2011) found that HULIS undergoes degradation under UV and ozone conditions, which in turn reduces its aromaticity and molecular weight. Thus, the higher $SUVA_{254}$ values in smaller particle in this study may be due to fresh WSOC in these particle size ranges. In contrast, WSOC in larger particles undergoes a series of degradation reactions during the aging process, resulting in a decrease in aromaticity/molecular weight. The average $MAE_{365}$ values of particles in the size range of <0.26 μm, 0.44–0.77 μm, 1.40–2.50 μm, and 2.50–10.0 μm were 0.6937, 0.4656, 0.4610, 0.2426 $m^2\ g^{-1}$, respectively, indicating that WSOC in smaller particles had stronger light absorption capabilities (Huang et al., 2022). This is because WSOC in smaller particles contained more chromophores, such as nitrogenous chromophores (see Section 3.1.2), resulting in stronger light absorption capabilities (Wu et al., 2018). The average AAE values were highest in particle sizes of <0.26 μm, suggesting the existence of more wavelength dependence light absorption properties in smaller particles (Wu et al., 2018)."

11. Lines 154-172: This part is confused, and should be rewritten.

**Response: Sorry for making such confusion.** We have thoroughly rewritten this part in Line 185-196 as follows:

"The light absorption and fluorescence properties of WSOC varied with particle size (Table 1). Higher $SUVA_{254}$ for particles of < 0.26 μm than other particle sizes highlighted the relatively higher aromaticity/molecular weight of WSOC in smaller particles (Cawley et al., 2013). Baduel et al. (2011) found that HULIS undergoes degradation under UV and ozone conditions, which in turn reduces its aromaticity and molecular weight. Thus, the higher $SUVA_{254}$ values in smaller particle in this study may be due to fresh WSOC in these particle size ranges. In contrast, WSOC in larger

particles undergoes a series of degradation reactions during the aging process, resulting in a decrease in aromaticity/molecular weight. The average $MAE_{365}$ values of particles in the size range of <0.26 μm, 0.44–0.77 μm, 1.40–2.50 μm, and 2.50–10.0 μm were 0.6937, 0.4656, 0.4610, 0.2426 $m^2$ $g^{-1}$, respectively, indicating that WSOC in smaller particles had stronger light absorption capabilities (Huang et al., 2022). This is because WSOC in smaller particles contained more chromophores, such as nitrogenous chromophores (see Section 3.1.2), resulting in stronger light absorption capabilities (Wu et al., 2018). The average AAE values were highest in particle sizes of <0.26 μm, suggesting the existence of more wavelength dependence light absorption properties in smaller particles (Wu et al., 2018)."

12. Line 158: It is written: "highlighting the relatively higher aromaticity of WSOC in small particles; on the other hand in line 156: "the molecular weight of WSOC was higher for particle sizes of 2.50–10.0 μm in summer and 1.40– 2.50 μm in winter", and (line 166/167): "It has been documented that more aromaticity and larger molecular sizes of light absorbing substances tend to have higher MAE365, which is mainly derived...." You concluded that the higher aromaticity is found in the smallest (below 0.26 μm), but on the other hand, higher molecular weight of WSOC was typical for bigger sizes.

**Response: Sorry for the confusion.** Previous study found that the $S_{275–295}$ of the high-molecular-weight dissolved organic matter (DOM) were generally lower than those for the low-molecular-weight (Helms et al., 2008). Therefore, we speculated that the molecular weight of WSOC in larger particles in this study might be higher. However, we neglected that the origins and formation mechanisms of aquatic DOM and aerosol WSOC are different, which implies that examining the molecular weight of aerosol WSOC based on our knowledge of $S_{275-295}$ values of aquatic DOM might be speculative. Therefore, we have removed $S_{275-295}$ from manuscripts. Additionally, $SUVA_{254}$ is an important parameter to characterize the aromaticity and molecular weight of WSOC (Voliotis et al., 2017; Zhang et al., 2022). In this study, we found that $SUVA_{254}$ were higher for particle sizes of <0.26 μm than other larger particle, suggesting the relatively

higher aromaticity/molecular weight of WSOC in smaller particles. This is consistent with the results from the $MAE_{365}$ analysis. We have made the following modifications in Line 185-196 according to your suggestion:

"The light absorption and fluorescence properties of WSOC varied with particle size (Table 1). Higher $SUVA_{254}$ for particles of < 0.26 μm than other particle sizes highlighted the relatively higher aromaticity/molecular weight of WSOC in smaller particles (Cawley et al., 2013). Baduel et al. (2011) found that HULIS undergoes degradation under UV and ozone conditions, which in turn reduces its aromaticity and molecular weight. Thus, the higher $SUVA_{254}$ values in smaller particle in this study may be due to fresh WSOC in these particle size ranges. In contrast, WSOC in larger particles undergoes a series of degradation reactions during the aging process, resulting in a decrease in aromaticity/molecular weight. The average $MAE_{365}$ values of particles in the size range of <0.26 μm, 0.44–0.77 μm, 1.40–2.50 μm, and 2.50–10.0 μm were 0.6937, 0.4656, 0.4610, 0.2426 $m^2\,g^{-1}$, respectively, indicating that WSOC in smaller particles had stronger light absorption capabilities (Huang et al., 2022). This is because WSOC in smaller particles contained more chromophores, such as nitrogenous chromophores (see Section 3.1.2), resulting in stronger light absorption capabilities (Wu et al., 2018). The average AAE values were highest in particle sizes of <0.26 μm, suggesting the existence of more wavelength dependence light absorption properties in smaller particles (Wu et al., 2018)."

Reference:

1. Helms J R, Stubbins A, Ritchie J D, et al. Absorption spectral slopes and slope ratios as indicators of molecular weight, source, and photobleaching of chromophoric dissolved organic matter[J]. Limnology and Oceanography, 2008, 53(3): 955-969.

2. Voliotis A, Prokeš R, Lammel G, et al. New insights on humic-like substances associated with wintertime urban aerosols from central and southern Europe: Size-resolved chemical characterization and optical properties[J]. Atmospheric Environment, 2017, 166: 286-299.

3. Zhang T, Huang S, Wang D, et al. Seasonal and diurnal variation of $PM_{2.5}$ HULIS

over Xi'an in Northwest China: Optical properties, chemical functional group, and relationship with reactive oxygen species (ROS)[J]. Atmospheric Environment, 2022, 268: 118782.

13. Line 181: ...that aged WSOC undergone a growth process with increasing particle size". This statement is a bit strange. Please, correct. Generally, with the process of growth the particle size increases.

**Response: Thank you very much for your valuable advice.** The corresponding revision has been provided in Line 204-206 as follows:

"The overall fluorescence intensity per TOC (FI/TOC) decreased steadily with increasing particle size (Table 1). The fluorescence intensity was higher for fresh than aged brown carbon (BrC), as previously observed by Fan et al. (2020). Therefore, the FI/TOC further highlights that WSOC may have undergone a cascade of aging processes (e.g., photochemical aging and oxidative aging) with increasing particle size, and thus has weaken its fluorescence intensity (Kuang et al., 2021; Wu et al., 2021)."

14. Line 194: Both samples (which samples did you have in mind)?

**Response: Sorry for our mistake.** It should be "all samples" instead of "both samples" here. We have rewritten this sentence in Lines 227 as follows:

"The FTIR spectra predominantly exhibited the presence of oxygen containing functional groups and aliphatic C–H groups for all samples (Duarte et al., 2005)."

15. Chapter 3.2.1: I suggest at least short concluding remarks from FTIR analyses.

**Response: We are grateful for the suggestion.** We have added a short summary from FTIR analysis in Line 239-243 as follows:

"It is obvious that smaller particles contained more oxygenated species and nitrogenous organic compounds than larger particles did. Nitrogenous organic compounds with different aromatic structures are considered to be important light absorbers (Wu et al., 2022), resulting in stronger light absorption of WSOC in smaller particles. This is consistent with the results from the optical absorption properties analysis discussed in

Section 3.1.1."

16. Lines 209-210: This part can be moved in the Introduction.

**Response: Thank you very much for your suggestion.** We have moved these sentences to Line 69-78 of the Introduction, and revised and supplemented it carefully as follows:

"The commonly used analytical methods to characterize the optical properties of WSOC are three-dimensional excitation-emission matrix (EEM) spectroscopy and ultraviolet–visible (UV–Vis) absorption spectroscopy (Zhang et al., 2021; Yang et al., 2020). The EEM spectroscopy is a rapid as well as informative method to identify chromophores that may not be distinguished by UV–Vis absorption spectroscopy (Chen et al., 2019; Xiao et al., 2020). Therefore, EEM spectroscopy has been widely applied in atmospheric WSOC characterization (Fu et al., 2015; Qin et al., 2018). However, such a technique has not been widely applied to investigate the fluorescence properties of WSOC in different particle sizes. Fourier transform infrared (FTIR) spectroscopy has been frequently used for the identification of WSOC functional groups (Chen et al., 2016a), although this analysis is difficult to perform quantitatively. pH titration enables qualitative and quantitative analyses of functional groups on the surface of substances (Zhang et al., 2011; Xiao et al., 2014), and this approach has recently been successfully applied to the characterization of WSOC in ambient $PM_{2.5}$ (Qin et al., 2021b), but not yet on size-resolved WSOCs."

17. Line 219: Not only phenol; so, it is better to say "aromatic compounds".

**Response: We are grateful for the suggestion.** We have corrected it in Line 256 as follows:

"It has been demonstrated that aromatic compounds (e.g., phenol) are abundant in biomass burning particles (Laskin et al., 2015; Lin et al., 2016; Sannigrahi et al., 2006)."

18. Lines 224/225: This is definitely not good enough / not a sufficient explanation. Just to remind, WSOC in winter (and summer) aerosols can be of primary and secondary

source. Please, check some more references and add appropriate explanation. For example, nitroaromatic compounds (phenolic like) can be largely formed via different reactions in gas or aqueous phase, and can be present in fine particles (below 1µm) in all seasons (See ref. e.g. Frka et al., Chemosphere 2022).

**Response: We are extremely grateful for pointing out this problem.** According to your suggestion, we have added the appropriate explanation and related literature in Line 254-261 as follows:

"In contrast, the contribution of (strong) phenolic groups was highest in smaller particles (< 0.77 µm) and lowest in larger particles (1.40–2.50 µm). This pattern indicates that the phenolic groups were abundant in WSOC of smaller particles. It has been demonstrated that aromatic compounds (e.g., phenol) are abundant in biomass burning particles (Laskin et al., 2015; Lin et al., 2016; Sannigrahi et al., 2006). Additionally, nitrophenols and their derivatives have been found to be possibly associated with the gas-phase oxidation of anthropogenic VOCs (Frka et al., 2022; Wang et al., 2019). Therefore, the size-distribution of acidic group clearly indicates that WSOC with sizes of < 0.77 µm mainly originated from biomass burning, although the contribution of secondary formation should not be completely neglected."

19. Line 232: Can you really say "monotonic"?

**Response: Sorry for making such confusion.** We have re-written this sentence in Line 272-273 as follows:

"As shown in Fig. 3, the absorption of WSOC gradually increased with increasing pH for almost all samples"

20. Line 236/237: Can you give some explanation?

**Response: We are grateful for the suggestion.** The absorption in particle sizes of < 0.26 µm and 0.44–0.77 µm in summer and < 0.26 µm and 2.50–10.0 µm in winter showed a more pronounced pH dependence. In addition, we found that the $MAE_{365}$ for particle sizes of < 0.26 µm changed more sharply with increasing pH than larger particles. It can be concluded that the chromophores in smaller particle show a more

pronounced pH dependence. Phillips et al. (2017) found that the variation of the light absorption properties of brown carbon (BrC) with pH is the result of structural changes in the acidic functional groups of chromophores (substituted nitro-aromatics and phenols). Therefore, the strong pH sensitivity of chromophores in smaller particles might be related to the fact that these samples have a larger proportion of aromatic species (e.g., nitrogenous aromatic species). This is consistent with the results from Table 1, where more aromatic species were found in smaller particles. Based on your suggestion, we have provided explanations for the pH-dependent behavior of the light absorption properties in Line 287-293 in the next Section, and we have made the following modification in Line 277 as follows:

Line 277: "The results suggest that the absorption in particle sizes of < 0.26 μm and 0.44–0.77 μm in summer and < 0.26 μm and 2.50–10.0 μm in winter showed a more pronounced pH dependence, as further illustrated below."

Line 287-293: "Overall, the average $MAE_{365}$ for particles of < 0.26 μm changed more sharply with increasing pH (with a 12.7% increase per pH unit) due to the stronger pH-dependence of the absorption spectra of smaller particles compared to the cases of larger particles, as described above. Phillips et al. (2017) found that the variations of the light absorption properties of BrC with pH was the result of structural changes in the nitro-aromatics and phenols. More aromatic species were found in the smaller than larger particles in this study (Table 1). Therefore, the strong dependence of light absorption properties on pH in smaller particles might be related to the greater content of aromatic species (e.g., nitrogenous aromatic species) in their WSOC."

Reference:

Phillips, S. M., et al., 2017. Light absorption by brown carbon in the Southeastern United States is pH-dependent. Environ. Sci. Technol. 51(12), 6782–6790. DOI: 10.1021/acs.est.7b01116.

21. Lines 243-246: Make two sentences.

**Response: We are grateful for the suggestion.** Based on your suggestion, we have

revised this sentence in Lines 298-301 as follows:

"Following the work of Dryer et al. (2008), the difference absorbance spectra ($\Delta$absorbance ($\lambda$)) can identify whether specific spectral bands defined as carboxyl and phenolic groups undergo significant changes within their corresponding pH ranges. Therefore, the $\Delta$absorbance ($\lambda$) was calculated in this study to examine the behavior of groups in resonance with chromophores in WSOC under pH titration."

22. Line 264: High pH values (above 7) are very rare for actual atmospheric conditions (clouds, fog, and aqueous aerosol), usually one can find more acidic conditions.

**Response: Thanks for your kind reminder.** You are right that, generally, the pH of most actual ambient particulate samples is acidic. In this study, we found an overall increasing trend of $MAE_{365}$ with increasing pH, indicating that WSOC would enhance its influence on climate as pH increases, not only for in the alkaline environment. We are very sorry that we were not describing it accurately here. According to your suggestion, we have carefully revised it in Line 286-287 as follows:

"The pH-dependent $MAE_{365}$ suggests the different climate effects WSOC may play under different pH conditions, the its climate impact would be enhanced as pH increases (Aiona et al., 2018)."

23. Lines 267/268: Please, give some explanation, why $MAE_{365}$ for the smallest particles changed more.

**Response: We gratefully appreciate for your comment.** Overall, the absorption and $MAE_{365}$ for particle sizes of < 0.26 μm changed more sharply with increasing pH than larger particles were observed, indicating that the chromophores in smaller particles show a more pronounced pH dependence. Phillips et al. (2017) found that the variation of the light absorption properties of brown carbon (BrC) with pH is the result of structural changes in the acidic functional groups of chromophores (substituted nitro-aromatics and phenols). Therefore, the strong pH sensitivity of chromophores in smaller particles might be related to the fact that they have a larger proportion of aromatic species (e.g., nitrogenous aromatic species). This is consistent with the results from

Table 1, where more aromatic species were found in smaller particles. Based on your suggestion, we have added some explanations for the strong dependence of the light absorption properties on pH in smaller particles in Line 287-293 as follows:

"Overall, the average $MAE_{365}$ for particles of < 0.26 μm changed more sharply with increasing pH (with a 12.7% increase per pH unit) due to the stronger pH-dependence of the absorption spectra of smaller particles compared to the cases of larger particles, as described above. Phillips et al. (2017) found that the variations of the light absorption properties of BrC with pH was the result of structural changes in the nitro-aromatics and phenols. More aromatic species were found in the smaller than larger particles in this study (Table 1). Therefore, the strong dependence of light absorption properties on pH in smaller particles might be related to the greater content of aromatic species (e.g., nitrogenous aromatic species) in their WSOC."

Reference:

Phillips, S. M., et al., 2017. Light absorption by brown carbon in the Southeastern United States is pH-dependent. Environ. Sci. Technol. 51(12), 6782–6790. DOI: 10.1021/acs.est.7b01116.

24. Figure 6 is not necessary here, since the results are present also in Fig.7.

**Response: Thank you very much for your valuable advice.** Figure 6 provides information on how pH affects the fluorescence intensity and peak position of WSOC. However, Figure 7 shows the variation of the average fluorescence intensity per unit TOC (FI/TOC) only. Therefore, it will be impossible to understand how the changes of fluorescence peak position with increasing pH if Figure 6 is deleted.

25. Line 281: The redshift and blueshift cannot be seen.

**Response: Sorry for our mistake here.** A redshift of fluorescence peak with increasing pH for the summer samples can be observed in Fig.6 instead of Fig. 7. We have corrected it in Line 324 as follows:

"Furthermore, a redshift of fluorescence peak positions with increasing pH was observed in summer (Fig. 6),"

26. Lines 282-285: You speculate that –COOH and –OH groups influence the most on fluorescence behavior. What about aromatic groups (line 282)?

**Response: Thank you very much for your question.** Sorry for our mistake in Line 282: the "aromatic groups" should be "acidic/basic groups in aromatic compounds". Based on Coble et al. (2014), it was found that the dissociation or protonation of acidic and basic groups in aromatic compounds typically leads to red and blue shifts of fluorophores. Therefore, we speculated that –COOH, –OH, and –NO$_2$ groups in aromatic compounds influence the fluorescence behavior of WSOC under pH titration. We have made the following modifications in Line 325-328 according to your suggestion:

"Protonation and dissociation of the acidic/basic groups in aromatic compounds can generally lead to a shift in fluorophores (Coble et al., 2014; Schulman et al., 1985). For example, the dissociation of the electron withdrawing groups (e.g., –COOH and –NO$_2$) leads to a blueshift in fluorophores, while the dissociation of the electron donating groups (e.g., –OH) results in a redshift in fluorophores (Schulman et al., 1985)."

Reference:

Coble, P., Lead, J., Baker, A., Reynolds, D. M., and Spencer, R.: Aquatic Organic Matter Fluorescence, Cambridge University Press, New York, USA, 2014.

27. Line 303: From Fig. 9, I would not make such a conclusion. It can be seen that for both, HULIS1 and HULIS2, $F_{max}$ decreased with pH very similarly (they had the same trend).

**Response: Thank you for your precious comments and advice.** The $F_{max}$ of different fluorescent components showed a similar variation pattern with increasing pH as you said. For example, the $F_{max}$ of all fluorescence components showed a peak at pH 3, and then tended to decrease with increasing pH. However, the magnitude of the effect of

pH on $F_{max}$ of different fluorescent components was different. For example, HULIS2 changed more sharply with increasing pH. We have corrected it in Line 349-354 according to your suggestion as follows:

"$F_{max}$ of different fluorescent components showed a similar variation pattern with increasing pH. For example, $F_{max}$ of all fluorescence components showed a peak at pH 3, and then tended to decrease with increasing pH. However, the magnitude of the effect of pH on $F_{max}$ of different fluorescent components was different. In contrast, HULIS2 varied significantly while HULIS1 and protein-like organic matter varied slightly with pH. The HULIS2 fluorophores showed the most susceptibility to acidity, likely indicative of a greater proportion of acidic/basic groups in HULIS2 fluorophores."

28. Lines 312-318: Not clear.

**Response: Thank you for pointing out this problem in the manuscript.** We have re-written this part in Line 359-366 as follows:

"AQY and Stokes shift were used to investigate the efficiency and energy change of the fluorescence process of WSOC. As shown in Fig. 10, AQY was also pH-dependent in all WSOC samples, and it generally decreased with increasing pH. This result is similar to a conclusion reported in our previous study (Qin et al., 2021b). It has been reported that a larger rate of non-radiative transition seems to be favorable for the AQY of fluorophores (Xiao et al., 2020). Thus, our AQY data presented here indicate the rate of non-radiative transition of the WSOC fluorophores decreased with increasing pH. Additionally, a high AQY of a substance means that only a small portion of the absorbed radiative energy will be converted into heat, thus reducing its heating effect (Aiona et al., 2018). Hence, the pH-dependent AQY presented above further confirms that the impact of WSOC on climate would be enhanced with increasing pH."

29. Lines 325/326: What can you say based on these results (what about aromatic compounds in winter WSOC samples)?

**Response: Thank you very much for your suggestion.** Xiao et al. (2019) found that the Stokes shifts are related to the π-conjugated system. Therefore, Stokes shifts in

winter samples exhibited strong pH dependence may be related to the high content of aromatic compounds (with extensive π-conjugated system). According to your suggestion, we have added a brief description for the seasonal difference of Stokes shifts in Lines 374-377 as follows:

"However, WSOC with lower pH tend to have higher intensity at high Stokes shift values (about 2.0 $\mu m^{-1}$). This trend was evident in winter samples, probably resulted from the high content of aromatic compounds in these samples (with extensive π-conjugated system) because pH had an important impact on the π-conjugated systems and thus changed Stokes shifts of WSOC (Xiao et al., 2019)."

Reference:

Xiao K, Han B, Sun J, et al. Stokes shift and specific fluorescence as potential indicators of organic matter hydrophobicity and molecular weight in membrane bioreactors[J]. Environmental Science and Technology, 2019, 53(15): 8985–8993.

30. Chapter 4. Change the title as: "Summary and atmospheric implications"

**Response: Thank you very much for your valuable advice.** We have changed the title of Section 4 to "Summary and atmospheric implications" in the revised manuscript as follows:

"**4 Summary and atmospheric implications**"

31. Lines 338/339: As I said above, this is definitely not good enough. Please, correct.

**Response: Thanks for your kind reminder.** We have modified this statement in Line 393-396 as follows:

"The carboxylic groups tend to be enriched in larger particles (1.40–2.50 μm), whereas the contribution of phenolic groups was highest in smaller particles (< 0.77 μm) and lowest in larger particles (1.40–2.50 μm), indicating that WSOC with sizes of < 0.77 μm was most likely derived from biomass burning, although the contribution from secondary formation source cannot be completely excluded."

32. Line 345: Please, see my comment above.

**Response: Thanks for your kind reminder.** We have made the following modifications in Line 399-401 according to your suggestion:

"The results of PARAFAC analysis showed that the different fluorescent components showed a similar variation pattern with increasing pH, but the HULIS2 fluorophores were most sensitive to acidity."

33. Figures: All figures/their subtitles are needed to be updated with missing information (see e.g. below for Fig.4).

Figure 1: Complete the information in the capture.

Figure 2: Complete the information in the capture.

Figure 3: Correct. In summer () and winter particles ()

Figure 4: Complete the information, for example as: Difference absorbance spectra (Δabsorbance) of WSOC in winter and summer particles of different sizes in the pH range......

Figure 5: Complete the information.

Figures 6, 7: Complete the information.

**Response: We are extremely grateful for pointing out these problems.** According to your suggestion, we have added the missing information to the titles of figures as follows:

"**Figure 1.** FTIR spectra of WSOC in particles of different sizes in summer and winter at raw pH.

**Figure 2.** The distribution of the acidic group of WSOC in particles of different sizes in summer and winter.

**Figure 3.** pH dependence of absorption per unit mass of WSOC in particles of different sizes in summer (upper row) and winter (lower row) in the pH range of 2–10.

**Figure 4.** pH dependence of mass absorption efficiency ($MAE_{365}$) for WSOC in particles of different sizes in (a) summer and (b) winter in the pH range of 2–10.

**Figure 5.** Difference absorbance spectra (Δabsorbance) of WSOC in particles of different sizes in winter and summer in the pH range of 2–10.

**Figure 6.** pH dependence of the FI$_m$/TOC for WSOC in particles of different sizes in summer (upper row) and winter (lower row) in the pH range of 2–10.

**Figure 7.** pH dependence of the FI/TOC for WSOC in particles of different sizes in (a) summer and (b) winter in the pH range of 2–10.

**Figure 9.** pH dependence of the $F_{max}$ of fluorescence components for WSOC in particles of different sizes in summer (upper row) and winter (lower row) in the pH range of 2–10.

**Figure 10.** pH dependence of apparent quantum yield (AQY) for WSOC in particles of different sizes in summer (upper row) and winter (lower row) in the pH range of 2–10.

**Figure 11.** pH dependence of the Stokes shift for WSOC in particles of different sizes in summer (upper row) and winter (lower row) in the pH range of 2–10.

34. Technical corrections

Line 199: Correct the sentence.

Line 211:...in the range of 3.0-9.0.

Line 312: not materials/ use another word

Line 347: Delete "an indication of"

**Response: We are extremely grateful for pointing out these problems.** We have made the following modifications in the revised manuscripts according to your suggestion:

Line 233-234: "A peak at 1641 cm$^{-1}$ was also previously reported, which was attributed to conjugated carbonyl (C=O) groups and aromatic rings (C=C) (Zhang et al., 2022)."

Line 249: "Figure 2 shows the distributions of the acidic groups of WSOC with a p$K_a$ in the range of 3.0–9.0."

Line 364: "Additionally, a high AQY of a substance means that only a small portion of the absorbed radiative energy will be converted into heat, thus reducing its heating effect (Aiona et al., 2018)."

Line 402-403: "The results presented in this study suggest that the chemical characteristics and optical properties of WSOC with different particle sizes can provide information on their sources and atmospheric aging processes."

---

## Editor Decision (ED1)

**Additional comments on acp-2022-321**

Based on the comments of the expert in the field and myself, and after my consideration, the manuscript is of adequate atmospheric interest to merit publication in Atmospheric Chemistry and Physics as a Measurement report. The authors have thoroughly responded all the questions/comments raised by the reviewer and me, and modified the manuscript according to the suggestions and important changes have been done, so that some confusions have been clarified.

However, I have still some additional comments, which are needed to be solved before publication.

**Comments/ errors:** (lines in the revised version of MS)

Line 53:  …wood burning was the most important….
Line 55: Correct as: …"that HULIS in smaller particles was likely derived from local sources, while in larger particles from secondary organic aerosols (SOA)…«
Line 101: Still not clear. "For each particle size, a quarter of each filter of all the collected samples in summer or winter season were mixed together« Do you mean that all quarters of filters of each size (summer/winter) were combined in one sample?

Lines 190/191: "The average MAE365 values of particles in the size range of <0.26 µm, 0.44–0.77 µm, 1.40–2.50 µm, and 2.50–10.0 µm were 0.6937, 0.4656, 0.4610, 0.2426 m2 g−1, respectively«
As I can see, these are the average values for summer and winter. Please, correct the sentence appropriately.

Line 194: …the average AAE values were the highest in…
Line 193: ...nitrogen chromophores

Line 255 (line 394): …was the highest in smaller particles (< 0.77 µm) and the lowest in larger particles…

Line 258: »…nitrophenols and their derivatives have been found to be possibly associated with the gas-phase oxidation of anthropogenic VOCs« Not only gas-phase, but may be also the result of aqueous-phase oxidation reactions.

Lne 286: This sentence should be changed as: »The pH-dependent MAE365 suggests that under different pH conditions WSOC may have different impact on climate (i.e., climate impact would be enhanced as pH increases)."

Line 290: … that the variations of the light absorption properties of BrC with pH were the result…

Lines 292/293: "…to the greater content of aromatic species (e.g., nitrogenous aromatic species) in their WSOC.
Better as: …to the higher content… (e.g. nitro-aromatic species) in WSOC.

Lines 322/323 (and elsewhere): Please round the %! (e.g. 3.79%=3.8%)
Line 404: "as pH increases« can be deleted (as you already say »with increasing pH)

---

## Author Response (AR2)

Dear Editor-in-Chief,

We greatly appreciate the editor for thoroughly examining our manuscript and providing very helpful comments to guide our revision. Here we submit a new version (No.: acp-2022-321), entitled: "**Measurement Report: Investigation of pH- and particle size-dependent chemical and optical properties of water-soluble organic carbon: implications for its sources and aging processes**". We have carefully addressed all the comments provided by the editor. In the attachment, an item-by-item response to the comments of the editor is given below. All revisions are highlighted in blue color in the main text of the revised manuscript.

Thank you for taking care of the review process for this paper.

Sincerely,

Prof. Jihua Tan and coauthors

College of Resources and Environment, University of Chinese Academy of Sciences, Beijing 100049

tanjh@ucas.ac.cn

**Editor's Comments**

Based on the comments of the expert in the field and myself, and after my consideration, the manuscript is of adequate interest to merit publication in Atmospheric Chemistry and Physics as a Measurement report. The authors have thoroughly responded all the questions/comments raised by the reviewer and me, and modified the manuscript according to the suggestions and important changes have been done, so that some confusions have been clarified.

However, I have still some additional comments, which are needed to be solved before publication.

**Response to the Editor's Comments:**

1. Comments/ errors: (lines in the revised version of MS)

**Response: Sorry for our mistake.** We have carefully checked the lines in this version of the revised manuscript.

2. Line 53: …wood burning was the most important….

**Response: Sorry for our mistake.** We have changed "were" to "was" according to your suggestion in Line 54, modified as shown below:

"Frka et al. (2018) found that wood burning was the most important source of humic-like substances (HULIS) in the aerosol accumulation mode (from ~0.1 to ~2 μm) during the autumn and winter;"

3. Line 55: Correct as: "…that HULIS in smaller particles was likely derived from local sources, while in larger particles from secondary organic aerosols (SOA)…"

**Response: We are grateful for your valuable advice.** According to your comment, the corresponding revision has been provided in Line 55-57 as follows:

"Jang et al. (2019) reported that HULIS in smaller particles was likely derived from local sources, while in larger particles from secondary organic aerosols (SOA) in the atmosphere,"

4. Line 101: Still not clear. "For each particle size, a quarter of each filter of all the collected samples in summer or winter season were mixed together. Do you mean that all quarters of filters of each size (summer/winter) were combined in one sample?

**Response: Sorry for making such confusion.** You are right that a quarter of all filters of each size (summer/winter) were mixed in a bottle and then ultrapure water was added to extract WSOC in this study. It has been revised in Line 102 and is now described as:

"A quarter of all filters of each size (summer/winter) were mixed together in a bottle, extracted twice via ultrasonication in Milli-Q water for 15 min to achieve the extensive release of solubilized WSOC,"

5. Lines 190/191: "The average $MAE_{365}$ values of particles in the size range of <0.26 μm, 0.44–0.77 μm, 1.40–2.50 μm, and 2.50–10.0 μm were 0.6937, 0.4656, 0.4610, 0.2426 m$^2$ g$^{-1}$, respectively". As I can see, these are the average values for summer and winter. Please, correct the sentence appropriately.

**Response: Thanks for your comments.** According to your suggestion, we have added the $MAE_{365}$ values of different particle sizes in summer and winter in Line 190-193 as follows:

"The $MAE_{365}$ values of particles in the size range of <0.26 μm, 0.44–0.77 μm, 1.40–2.50 μm, and 2.50–10.0 μm were 0.1258, 0.1321, 0.1014, and 0.1145 m$^2$ g$^{-1}$ in summer, respectively, and 1.2615, 0.7991, 0.8206, and 0.3707 m$^2$ g$^{-1}$ in winter, respectively, indicating that WSOC in smaller particles had stronger light absorption capabilities (Huang et al., 2022)."

6. Line 194: …the average AAE values were the highest in…

**Response: Thanks for the helpful comments.** According to your suggestion, we have revised this sentence in Line 195 as follows:

"The average AAE values were the highest in particle sizes of <0.26 μm,"

7. Line 193: ...nitrogen chromophores

**Response: Thanks for your reminder.** The corresponding revision has been provided

in Line 194 as follows:

"such as nitrogen chromophores (see Section 3.1.2),"

8. Line 255 (line 394): …was the highest in smaller particles (< 0.77 μm) and the lowest in larger particles…

**Response: Thank you very much for your valuable suggestion.** We have checked the whole text and made the following modifications in the revised manuscript:

Line 253-254: "the contribution of (strong) phenolic groups was the highest in smaller particles (< 0.77 μm) and the lowest in larger particles (1.40–2.50 μm)."

Line 282: "However, the $MAE_{365}$ for particles < 0.26 μm in winter, exhibited the highest value at pH 4,"

Line 385-386: "whereas the contribution of phenolic groups was the highest in smaller particles (< 0.77 μm) and the lowest in larger particles (1.40–2.50 μm),"

9. Line 258: "…nitrophenols and their derivatives have been found to be possibly associated with the gas-phase oxidation of anthropogenic VOCs". Not only gas-phase, but may be also the result of aqueous-phase oxidation reactions.

**Response: Thanks for your comments.** We have revised it in Line 257-258 according to your suggestions, and the details are as follows:

"nitrophenols and their derivatives have been found to be possibly associated with the gas-phase oxidation of anthropogenic VOCs and aqueous-phase oxidation processes in polluted high-$NO_x$ environments (Frka et al., 2022; Wang et al., 2019)."

10. Line 286: This sentence should be changed as: "The pH-dependent $MAE_{365}$ suggests that under different pH conditions WSOC may have different impact on climate (i.e., climate impact would be enhanced as pH increases)."

**Response: Thanks for your kind reminder.** We have made the following modifications in Line 284-285 according to your suggestion:

"The pH-dependent $MAE_{365}$ suggests that under different pH conditions WSOC may have a different impact on climate (i.e., climate impact would be enhanced as pH

increases) (Aiona et al., 2018)."

11. Line 290: … that the variations of the light absorption properties of BrC with pH were the result…

**Response: Thank you very much for your suggestion.** We have corrected it in Line 287-289 according to your suggestion:

"Phillips et al. (2017) found that the variations of the light absorption properties of BrC with pH were the result of structural changes in the nitro-aromatics and phenols."

12. Lines 292/293: "…to the greater content of aromatic species (e.g., nitrogenous aromatic species) in their WSOC. Better as: …to the higher content… (e.g. nitro-aromatic species) in WSOC.

**Response: Thanks for your reminder.** We have revised it in Line 290-291 as follows:

"the strong dependence of light absorption properties on pH in smaller particles might be related to the higher content of aromatic species (e.g., nitro-aromatic species) in WSOC."

13. Lines 322/323 (and elsewhere): Please round the %! (e.g. 3.79%=3.8%).

**Response: Thanks for your kind reminder.** We have checked the whole text and rounded all the % in Lines 318-320 and elsewhere.

Line 271-273: "On average, the absorbance for particle sizes of < 0.26 μm, 0.44–0.77 μm, 1.40–2.50 μm, and 2.50–10.0 μm increased by 4.6 %, 1.3 %, 0.6 %, and 0.9 %, respectively, per unit pH increase in summer, and by 1.3 %, 0.5 %, 0.5 %, and 2.9 %, respectively, in winter."

Line 318-320: "On average, the FI/TOC of < 0.26 μm, 0.44–0.77 μm, 1.40–2.50 μm, and 2.50–10.0 μm decreased by 3.8 %, 3.5 %, 4.7 %, and 6.8 %, respectively, per unit pH increase in winter, which are significantly more than those (0.6 %, 1.7 %, 0.2 %, and 2.5 %, respectively) in summer."

14. Line 404: "as pH increases" can be deleted (as you already say "with increasing

pH").

**Response: Thanks for your valuable comments.** We have deleted "as pH increases" from this sentence in Line 395-396 as follows:

"The variation of both MAE$_{365}$ and AQY of WSOC with increasing pH suggested the enhanced impact of WSOC on climate."